# Comprehensive analysis of the potential effect and mechanism of pyroptosis-related genes in treatment-related myeloid tumors

Jing Cheng[1‡], Weiyue Fang[2‡], Hongxia Tan[1‡], Xiaoxia Zhan[1]*, Xiaohui Zhu[2,3]*

1 Department of Laboratory Medicine, The First Affiliated Hospital of Sun Yat-sen University, Guangzhou, China, 2 School of Pharmacy, Shenzhen University Medical School, Shenzhen University, Shenzhen, China, 3 College of Pharmacy, Shenzhen Technology University, Shenzhen, China

‡ JC and WF and HT also contributed equally to this work.
* xiaoxiazhan14@163.com (XZ); zhuxiaohui@sztu.edu.cn (XZ)

## Abstract

Treatment-related myeloid neoplasms (t-MN) represent a severe complication of cancer therapy, characterized by poor prognosis and limited treatment options. This study presents a preliminary, exploratory bioinformatic analysis aimed at characterizing the expression landscape and potential regulatory roles of pyroptosis-related genes (PRGs) in a murine model of t-MN. Utilizing RNA-seq data (GEO: GSE135866), differential expression analysis identified 1286 DEGs. Cross-referencing 367 curated mouse PRGs revealed 46 pyroptosis-related DEGs (PRDEGs). Functional enrichment analysis (GO, KEGG) showed these PRDEGs are significantly involved in autophagy, inflammatory regulation, apoptosis, NOD-like receptor signaling, and the AMPK pathway. GSEA associated the broader gene set with PI3K-Akt and Notch signaling. Protein-protein interaction network analysis identified five critical hub genes: *Trp53*, *Mtor*, *Gpx3*, *Foxo3*, and *Cybb*. ROC curve analysis confirmed these hub genes exhibit significant differential expression and high diagnostic accuracy (AUC > 0.9) in distinguishing t-MN from controls. Furthermore, immunoinfiltration analysis (CIBERSORT) revealed significant differences in immune cell composition between t-MN and control samples and identified notable correlations between hub gene expression and specific immune cell abundances. Importantly, given the limited sample size and the use of murine bone marrow data, the statistical findings should be interpreted strictly at the exploratory and hypothesis-generating level. This study does not support definitive biological conclusions or causal inferences but rather aims to delineate the pyroptosis-related molecular profile in a preclinical t-MN model. The results are intended to inform and guide future investigations—including validation in larger cohorts, independent experimental models, and human clinical samples—to assess the translational potential of these candidate biomarkers and therapeutic targets.

**Data availability statement:** All relevant data for this study are publicly available from the GitHub repository (https://github.com/Sztu00879/Supporting-materials-for-t-AML.git).

**Funding:** This work was supported by the National Natural Science Foundation of China (Grant No. 81901659).

**Competing interests:** The authors have declared that no competing interests exist.

## Introduction

Therapy-related myeloid neoplasms (t-MNs), encompassing therapy-related myelodysplastic syndrome (t-MDS) and therapy-related acute myeloid leukemia (t-AML), represent severe complications arising after cytotoxic chemotherapy and/or radiotherapy [1,2]. The incidence of t-MNs has shown a steady increase in recent years [3]. Epidemiological studies have shown that this occurs in a notable proportion of patients who have received prior cancer treatment. Accounting for 10–20% of all MDS/AML cases, these malignancies are characterized by a dismal prognosis, with a median survival of less than 12 months compared with their de novo counterparts [4]. In the United States, the incidence of t-MNs is approximately 0.13 cases per 100,000 population. The two-year overall survival rates varied significantly by age, ranging from 51.3% in the 20–39 age group to 0% in those aged 80 years and above. The prognosis of patients with t-MN is poor, with high mortality rates [5]. Established risk factors for t-MN development include the type and intensity of prior therapy, patient age at initial treatment, and genetic predisposition [6]. Patients who have undergone alkylating agent-based chemotherapy or radiotherapy are at particularly high risk. Additionally, certain genetic mutations and polymorphisms may further increase the risk of developing t-MN, emphasizing the complex interactions between treatment and genetic factors in its development [7,8].

The pathogenesis of t-MNs involves multifactorial mechanisms and a series of molecular and cellular events [9]. Chemotherapy- and radiotherapy-induced DNA damage caused by chemotherapy and radiotherapy is a key initiating factor [10,11]. Such damage can lead to genomic instability, mutations, and chromosomal abnormalities. Specific genetic alterations—including mutations in genes involved in DNA repair (e.g., *Trp53*), epigenetic regulation, and signaling pathways—are frequently observed in t-MNs [12,13]. These genetic changes disrupt normal hematopoiesis and promote the clonal expansion of abnormal myeloid cells. Moreover, alterations in the bone marrow microenvironment, including changes in cytokine profiles and stromal cells, may contribute to the development and progression of t-MNs [14]. The cumulative effects of these genetic and microenvironmental alterations led to the emergence of a malignant clone characterized by dysregulated proliferation and differentiation.

The management of t-MNs presents several challenges. Patients with t-MNs generally have a worse prognosis than those with de novo myeloid neoplasms. A significant difficulty arises from underlying comorbidities resulting from previous cancer treatments, which limit the options and tolerability of therapeutic interventions, such as chemotherapy regimens [15]. Chemotherapy regimens that are effective in de novo cases might not yield similar results in t-MN, and the associated toxicities are often more severe, necessitating careful evaluation [16,17]. Allogeneic hematopoietic stem cell transplantation (HSCT) is considered a potentially curative option; however, its application is frequently restricted by factors such as age, comorbidities, and donor availability. Recent studies have highlighted the role of pre-existing clonal hematopoiesis in predisposing individuals to t-MN, particularly those with mutations in *DNMT3A* or *TET2* [18]. Furthermore,

relapse rates after HSCT remain high [19]. The heterogeneity of genetic and clinical features of t-MNs, which complicates the development of standardized treatment strategies [20]. Therapy resistance is a prevalent problem, and the mechanisms underlying this resistance are not fully understood [1]. Furthermore, accurate diagnosis and risk stratification of t-MNs could be challenging. Distinguishing t-MNs from other myeloid disorders with similar manifestations can be difficult, especially in cases with an unclear history of prior therapy [21]. The prognostic significance of various genetic alterations and biomarkers in t-MNs is still being defined, thereby impeding individualized treatment decisions [22].

Given the complexity and challenges associated with t-MNs, further research is urgently required. This study aimed to enhance the understanding of the epidemiology, pathogenesis, and clinical management of t-MNs. To address these knowledge gaps, we focused on pyroptosis-related differentially expressed genes (PRDEGs) in t-MNs. Previous research has established associations between pyroptosis and the development and progression of various cancers [23,24], suggesting a potential role for pyroptosis in t-MN pathogenesis. This potential link underscores the novelt and significance of our investigation. By comparing gene expression profiles between t-MN and control samples, we identified 46 PRDEGs associated with cell pyroptosis. Subsequent Gene Ontology (GO), Kyoto Encyclopedia of Genes and Genomes (KEGG) pathway enrichment, and gene set enrichment analysis (GSEA) revealed enrichment of these 46 genes in several key pathways, including NOD receptor, AMPK, PI3K-Akt, Notch, and Wnt signaling pathways. Receiver operating characteristic (ROC) curve analysis demonstrated that the five hub genes (*Trp53*, *Cybb*, *Foxo3*, *Mtor*, and *Gpx3*) exhibited high diagnostic accuracy (AUC > 0.9) in distinguishing t-MN from control samples. Immunoinfiltration analysis further revealed a significant positive correlation between the hub gene *Cybb* and neutrophils ($r > 0.0$, $p < 0.05$), and a significant negative correlation with resting NK cells ($r < 0.0$, $p < 0.05$). Collectively, these findings will contribute to an improved understanding of t-MN pathogenesis and may inform the development of novel therapeutic strategies, ultimately enhancing patient prognosis and quality of life.

t-MN represents a distinct clinical entity with unique epidemiological, pathogenic, and therapeutic characteristics [1]. Addressing the current therapeutic challenges requires a collaborative approach that integrates molecular pathology, clinical hematology, and translational research. This study represents a significant step toward bridging knowledge gaps and propelling the field of t-MN forward, contributing to the development of more effective diagnostic and therapeutic strategies for patients with t-MN.

## Methods (Fig 1)

### Data download

Using the "GEOquery" R package [25] (version 2.70.0) from the Gene Expression Omnibus (GEO) database [26] (https://www.ncbi.nlm.nih.gov/geo/), we downloaded the t-MN dataset GSE135866 [27]. The dataset GSE135866 was derived from *Mus musculus* and bone marrow tissues. The chip platform of dataset GSE135866 was GPL21103, and specific information is provided in Table 1. The t-MN and control samples were combined to integrate a GEO dataset (combined GEO dataset). The dataset comprised six t-MN samples and three control samples, all of which were included in this study.

Pyroptosis-related genes (PRGs) were obtained using the GeneCards database (https://www.genecards.org/) [28], which provides comprehensive information on human genes. We employed the keyword "Pyroptosis" in our search and retained only those PRGs classified as "protein coding" with a relevance score greater than 1. This process yielded 436 PRGs. Additionally, we searched for "Pyroptosis" as a keyword in the PubMed database (https://pubmed.ncbi.nlm.nih.gov/) from published literature [29–37], resulting in an additional 62 PRGs. A total of 455 PRGs were obtained after combining and deduplicating these datasets, and 367 PRGs were obtained after the phenotypic genes were converted into mouse genes. Further details are presented in S1 Table.

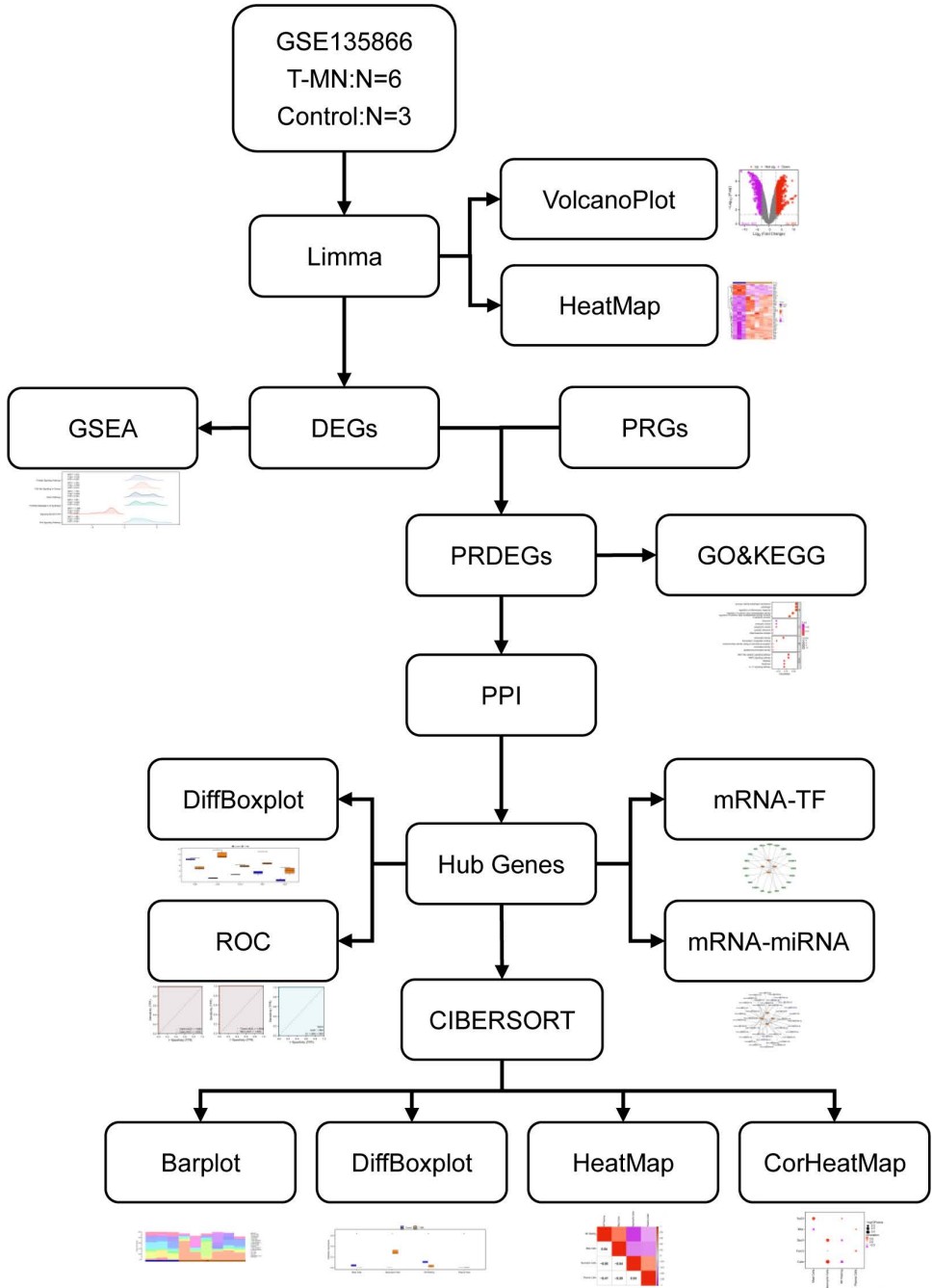

**Fig 1. Flow chart for the comprehensive analysis of PRDEGs.** t-MN, therapy-related myeloid neoplasm; GSEA, gene set enrichment analysis; DEGs, differentially expressed genes; PRGs, pyroptosis-related genes; KEGG, Kyoto Encyclopedia of Genes and Genomes; GO, Gene Ontology; PRDEGs, pyroptosis-related differentially expressed genes; PPI, protein–protein interaction; TF, transcription factor; ROC, receiver operating characteristic; DiffBoxplot, differential boxplot.

**Table 1. GEO microarray chip information.**

|  | GSE135866 |
| --- | --- |
| Platform | GPL21103 |
| Species | *Mus musculus* |
| Tissue | Bone marrow |
| Samples in the t-MN group | 6 |
| Samples in the control group | 3 |
| Reference | PMID: 32924016 |

GEO, Gene Expression Omnibus; t-MN, therapy-related myeloid neoplasm.

## DEGs related to pyroptosis PRDEGs in treatment-related myeloid tumors

The samples were categorized into t-MN and control samples based on the grouping provided in the combined GEO dataset. The "limma" R package [38] (version 3.58.1) was used to analyze gene expression differences between the t-MN and control samples. A threshold of |logFC|>3 and adjusted $p<0.05$ was established for identifying differentially expressed genes (DEGs). Genes with logFC>3 and adjusted $p<0.05$ were classified as upregulated, while genes with logFC<−3 and adjusted $p<0.05$ were categorized as downregulated. The results of the differential analysis were visualized using the "ggplot2" R package (version 3.4.4).

To obtain PRDEGs associated with t-MN, we conducted an intersection analysis of all DEGs with |logFC|>3 and adjusted $p<0.05$, along with PRGs obtained by difference analysis from the combined GEO dataset. Subsequently, a Venn diagram was generated to visualize the overlapping genes. PRDEGs were obtained, and a heatmap was created using the "pheatmap" R package (version 1.0.12).

## GO and KEGG pathway enrichment analyses

GO analysis [39] is a common method for conducting large-scale functional enrichment studies, including biological process (BP), cellular component (CC), and molecular function (MF). The KEGG [40] serves as a comprehensive database that stores information regarding genomes, biological pathways, diseases, and drugs, among other categories. We used the "clusterProfiler" R package [41] (version 4.10.0) to perform GO and KEGG pathway enrichment analyses of PRDEGs. The selection criteria were adjusted $p<0.05$, and false discovery rate (FDR) ($q$) < 0.25.

## GSEA

GSEA [42] was performed to assess the distribution trend of genes within predefined gene sets in the gene list ranked by phenotypic correlation to evaluate their contribution to phenotypes. Genes from the combined GEO dataset were sequenced based on their logFC values. Subsequently, GSEA was conducted on all genes within the combined GEO dataset using the "clusterProfiler" R package [41] (version 4.10.0). The parameters used in GSEA were as follows: the seed for permutation was 2020, and the number of calculations was 1000, with a minimum of 10 and a maximum of 500 genes per gene set. The gene sets of "c2.Cp.All. V2022.1. Hs Symbols" were acquired from the Molecular Signatures Database (MSigDB) [43]. GSEA was applied to detect enriched pathways of the DEGs with adjusted $p<0.05$ and FDR value ($q$) < 0.25 with consideration of significant enrichment.

## Protein–protein interaction (PPI) network and hub gene screening

A PPI network is a complex system comprising proteins that interact with each other. The STRING database serves as a resource for identifying known and predicted interactions between proteins. In this study, we utilized the STRING [44] database based on PRDEGs with a correlation coefficient greater than 0.4 (with the minimum required interaction score

of low confidence set at 0.150) to construct a PPI network associated with PRDEGs. In addition, five algorithms from the cytoHubba [45] plugin of Cytoscape [46] software were applied: maximal clique centrality (MCC), degree, maximum neighborhood component (MNC), edge percolated component (EPC), and closeness. Initially, we calculated the scores of the PRDEGs within the PPI network and subsequently ranked the top 10 PRDEGs based on these scores for selection. Finally, we performed an intersection analysis of the genes identified by the five algorithms and illustrated the results using a Venn diagram. Genes that intersected the algorithms were identified as hub genes related to focal death.

### Differential expression verification of hub genes and ROC curve analysis

To further explore the expression differences in hub genes between the t-MN and control samples in the integrated GEO datasets, a comparative grouping map was generated based on the expression levels of these hub genes. Next, the "pROC" R package [47] (version 1.18.5) was used to plot the ROC curve for the hub genes and to calculate the AUC. This analysis was conducted to assess the diagnostic potential of hub genes expression in t-MN development. The AUC of the ROC curve typically ranges from 0.5 to 1, with values closer to 1 indicating a superior diagnostic capability. AUC values of 0.5–0.7, 0.7–0.9, and > 0.9 suggest low, moderate, and high accuracy, respectively.

### Construction of regulatory network

The transcription factor (TF) controls gene expression at the post-transcriptional level by interacting with PRDEGs. The ChIPBase database [48] (http://rna.sysu.edu.cn/chipbase/) was used to retrieve the transcription factors, followed by analysis of their regulatory functions concerning genetic variations associated with pyroptosis PRDEGs. Cytoscape [45] was used to visualize the mRNA-TF regulatory network.

Additionally, miRNAs play important regulatory roles in biological development and evolution. miRNAs can regulate multiple target genes, and the same target gene can also be regulated by multiple miRNAs. To analyze the relationship between PRDEGs and miRNAs, miRNAs associated with PRDEGs were obtained from the TarBase [49] database (http://www.microrna.gr/tarbase), and the mRNA-miRNA regulatory network was visually analyzed using Cytoscape software.

### Immunoinfiltration analysis for treatment-associated myeloid tumors (CIBERSORT)

CIBERSORT [50] is a deconvolution tool for a transcriptomic expression matrix based on the principle of linear support vector regression to estimate the composition and abundance of immune cells in mixed cell populations. The CIBERSORT algorithm was used in conjunction with the LM22 characteristic gene matrix to filter data with immune cell enrichment scores greater than zero. This process yielded a specific immune cell infiltration matrix from the integrated GEO dataset, and a proportion histogram was generated for visualization. Subsequently, the "ggplot2" R package (version 3.4.4) was used to create a group comparison graph to show the differential expression of LM22 immune cells between the t-MN and control samples in the combined GEO dataset. Immune cells that exhibited significant differences between the two groups were selected for further analysis. The Spearman algorithm was applied to evaluate the correlation between immune cells and the correlation between hub genes and immune cells, retaining results with $p < 0.05$. The R package pheatmap (version 1.0.12) was used to generate a correlation heatmap to display the correlation analysis among immune cells, whereas the R package ggplot2 (version 3.4.4) was used to create a correlation bubble map to illustrate the correlation analysis between hub genes and immune cells.

### Statistical analysis

All data processing and analyses were performed using R software (version 4.2.2). Unless otherwise specified, the statistical significance of normally distributed continuous variables between the two groups was estimated using independent Student's *t*-test. Differences between non-normally distributed variables were analyzed using the

Mann–Whitney U test and Wilcoxon rank-sum test. Comparisons involving three or more groups were conducted using the Kruskal–Wallis test. Spearman's correlation analysis was performed to calculate the correlation coefficients among the different molecules. If not specified, all statistical $p$ values were bilateral, with $p < 0.05$ being considered statistically significant.

## Results

### PRDEGs in cells associated with treatment-related myeloid tumors

The integrated GEO dataset was categorized into the t-MN and control (untreated) samples. We conducted differential expression analysis using the "limma" R package to compare gene expression between the t-MN and control samples in the integrated GEO dataset. The analysis identified 1286 DEGs in the integrated GEO dataset meeting the threshold of $|logFC| > 3$ and adjusted $p < 0.05$, with 863 upregulated genes ($logFC > 3$ and adjusted $p < 0.05$) and 423 downregulated genes ($logFC < -3$ and adjusted $p < 0.05$). Based on the results of the differential expression analysis, a volcano map was generated (Fig 2A).

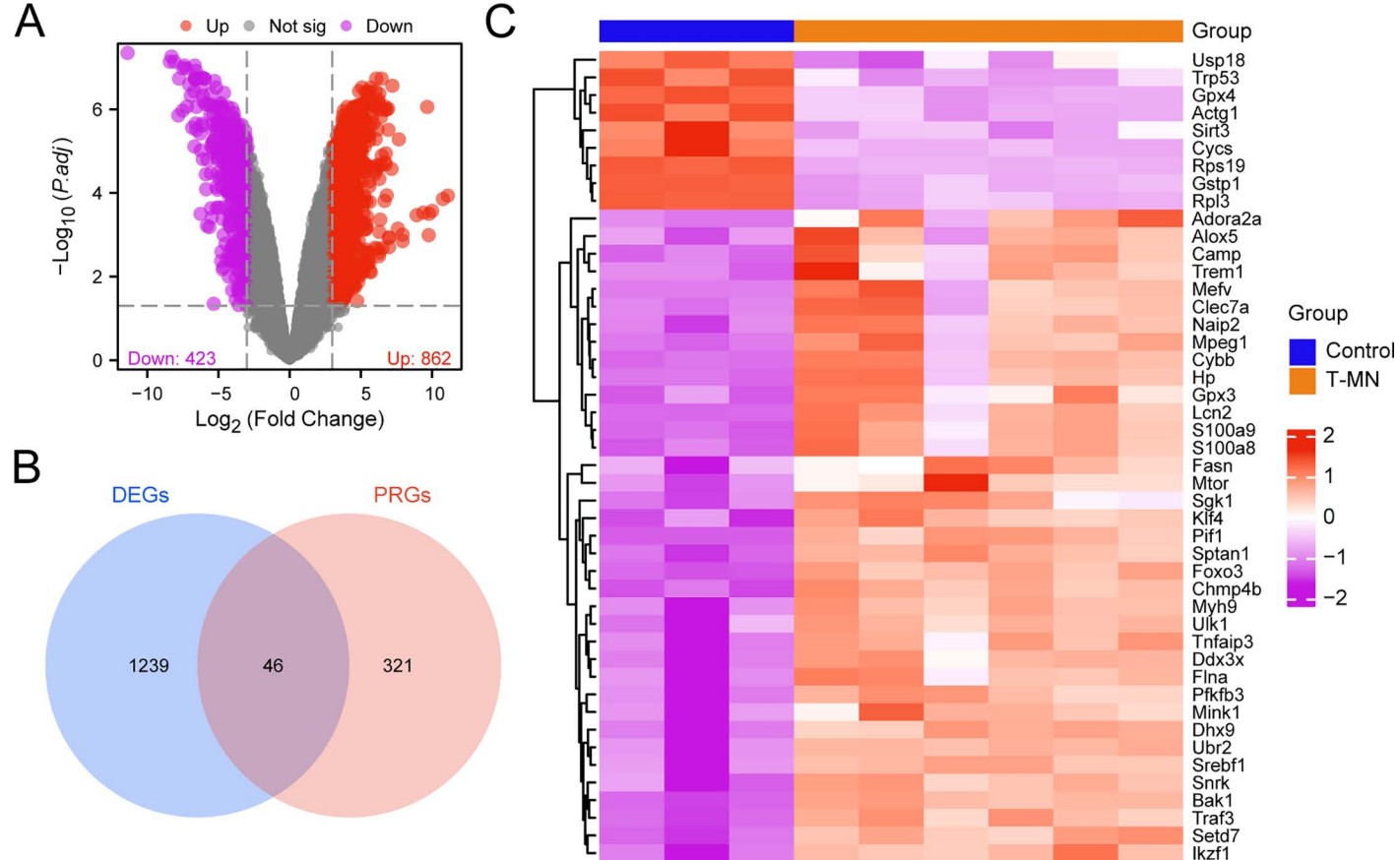

**Fig 2. Differential gene expression analysis. A.** Differential gene expression analysis using a volcano map of the t-MN and control samples in the integrated GEO dataset (combined GEO dataset). **B.** Venn diagram of DEGs and PRGs in the combined GEO dataset. **C.** Heat map of PRDEGs in the combined GEO dataset. t-MN, therapy-related myeloid neoplasm; DEGs, differentially expressed genes; PRGs, pyroptosis-related genes; PRDEGs, pyroptosis-related differentially expressed genes. In the heat map, orange represents the t-MN sample, whereas dark blue denotes the control sample. High expression is indicated in red, whereas low expression is represented in purple.

To obtain PRDEGs, the intersection of all DEGs with |logFC|>3 and adjusted $p<0.05$ and the PRGs were plotted, and Venn diagrams (Fig 2B) were created. We identified 46 PRDEGs including *Bak1*, *Pif1*, *Sptan1*, *Setd7, Foxo3*, and *Chmp4b*. Based on these intersection results, we further analyzed the differences in the expression of PRDEGs across different sample groups in the integrated GEO dataset and visualized the findings using heat maps generated using the "pheatmap" R package (Fig 2C).

## GO and KEGG pathway enrichment analyses

Through GO and KEGG pathway enrichment analyses, the relationships between BP, CC, MF, biological pathway (KEGG), and t-MNs were further explored in the 46 PRDEGs. GO and KEGG pathway enrichment analyses were performed using these 46 cellular PRDEGs (Table 2).

The analysis revealed that 46 PRDEGs were enriched in various BPs, including autophagy, regulation of inflammatory responses, cysteine-type endopeptidase activity, and apoptotic processes. Enriched CCs included ribosomes, endocytic vesicles, phagocytic vesicles, cytosolic ribosomes, and inflammasome complexes. The MFs encompassed antioxidant activity, transcription co-regulator binding, peroxide oxidoreductase activity, peroxidase activity, and glutathione peroxidase activity. Additionally, the analysis showed enrichment in several biological pathways, including NOD-like receptor, AMPK, measles, apoptosis, and IL-17 signaling pathways. The findings of the GO and KEGG pathway enrichment analyses are shown in a bubble map (Fig 3A). Following this analysis, network maps depicting the BP, CC, MF, and KEGG pathways were generated (Fig 3B–3E). Notably, the size of each node corresponded to the number of molecules associated with their respective entries.

**Table 2. Results of GO and KEGG pathway enrichment analyses for PRDEGs.**

| Ontology | ID | Description | GeneRatio | BgRatio | *p* | Adjusted *p* |
|---|---|---|---|---|---|---|
| BP | GO:2000116 | Regulation of cysteine-type endopeptidase activity | 9/46 | 240/28814 | 1.4e-10 | 1.93e-07 |
| BP | GO:0050727 | Regulation of inflammatory response | 10/46 | 351/28814 | 1.75e-10 | 1.93e-07 |
| BP | GO:0043281 | Regulation of cysteine-type endopeptidase activity involved in the apoptotic process | 8/46 | 210/28814 | 1.43e-09 | 1.05e-06 |
| BP | GO:0006914 | Autophagy | 10/46 | 464/28814 | 2.58e-09 | 1.14e-06 |
| BP | GO:0061919 | Process utilizing the autophagic mechanism | 10/46 | 464/28814 | 2.58e-09 | 1.14e-06 |
| CC | GO:0061702 | Inflammasome complex | 3/46 | 22/28739 | 5.79e-06 | 0.0011 |
| CC | GO:0045335 | Phagocytic vesicle | 4/46 | 113/28739 | 3.26e-05 | 0.0030 |
| CC | GO:0030139 | Endocytic vesicle | 4/46 | 211/28739 | 0.0004 | 0.0224 |
| CC | GO:0005840 | Ribosome | 4/46 | 236/28739 | 0.0006 | 0.0257 |
| CC | GO:0022626 | Cytosolic ribosome | 3/46 | 111/28739 | 0.0008 | 0.0281 |
| MF | GO:0016209 | Antioxidant activity | 6/45 | 84/28275 | 4.26e-09 | 1.01e-06 |
| MF | GO:0004602 | Glutathione peroxidase activity | 3/45 | 26/28275 | 9.55e-06 | 0.0011 |
| MF | GO:0001221 | Transcription coregulator binding | 4/45 | 132/28275 | 5.83e-05 | 0.0046 |
| MF | GO:0004601 | Peroxidase activity | 3/45 | 55/28275 | 9.33e-05 | 0.0055 |
| MF | GO:0016684 | Oxidoreductase activity, acting on peroxide as an acceptor | 3/45 | 59/28275 | 0.0001 | 0.0055 |
| KEGG | mmu04152 | AMPK signaling pathway | 6/36 | 127/9000 | 9.66e-06 | 0.0014 |
| KEGG | mmu04657 | IL-17 signaling pathway | 5/36 | 93/9000 | 3.09e-05 | 0.0023 |
| KEGG | mmu04621 | NOD-like receptor signaling pathway | 6/36 | 213/9000 | 0.0002 | 0.0067 |
| KEGG | mmu04210 | Apoptosis | 5/36 | 136/9000 | 0.0002 | 0.0067 |
| KEGG | mmu05162 | Measles | 5/36 | 146/9000 | 0.0003 | 0.0067 |

GO, Gene Ontology; BP, biological process; CC, cellular component; MF, molecular function; KEGG, Kyoto Encyclopedia of Genes and Genomes; PRDEGs, pyroptosis-related differentially expressed genes.

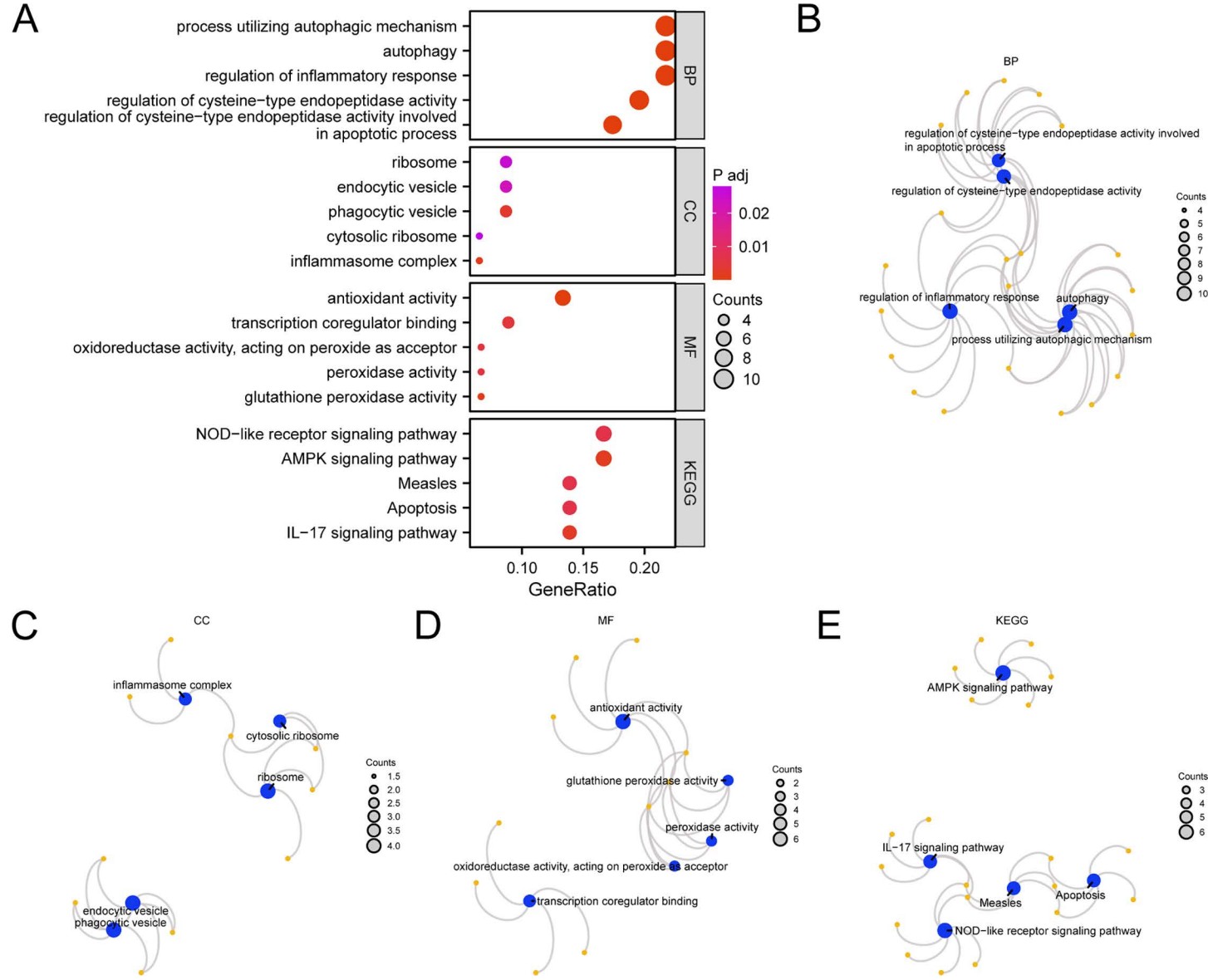

**Fig 3. GO and KEGG pathway enrichment analyses of PRDEGs. A.** The results of GO and KEGG pathway enrichment analyses for PRDEGs related to pyroptosis are shown. Ordinates represent GO and KEGG terms. B–E. The results of GO and KEGG pathway enrichment analyses for PRDEGs related to focal death are shown in the network diagram: BP **(B)**, CC **(C)**, MF **(D)**, and KEGG **(E)**. Blue nodes represent entries, orange nodes represent molecules, and lines represent the relationships between entries and molecules. PRDEGs, pyroptosis-related differentially expressed genes; GO, Gene Ontology; KEGG, Kyoto Encyclopedia of Genes and Genomes; BP, biological process; CC, cellular component; MF, molecular function. In the bubble diagram, the bubble size represents the number of genes, whereas the bubble color reflects the adjusted $p$ value. A redder hue indicates a smaller adjusted $p$ value, whereas a purple hue indicates a larger adjusted $p$ value. The screening criteria for the GO and KEGG pathway enrichment analyses were adjusted $p < 0.05$ and FDR value (**$q$**) < 0.25.

## GSEA

GSEA was performed to assess the influence of gene expression levels on t-MNs in the combined GEO dataset. This analysis examined the interrelationships among gene expression, BP, CC, and MF (Fig 4A). The detailed results are presented in Table 3. The findings indicated that all genes in the combined GEO dataset were significantly enriched in the

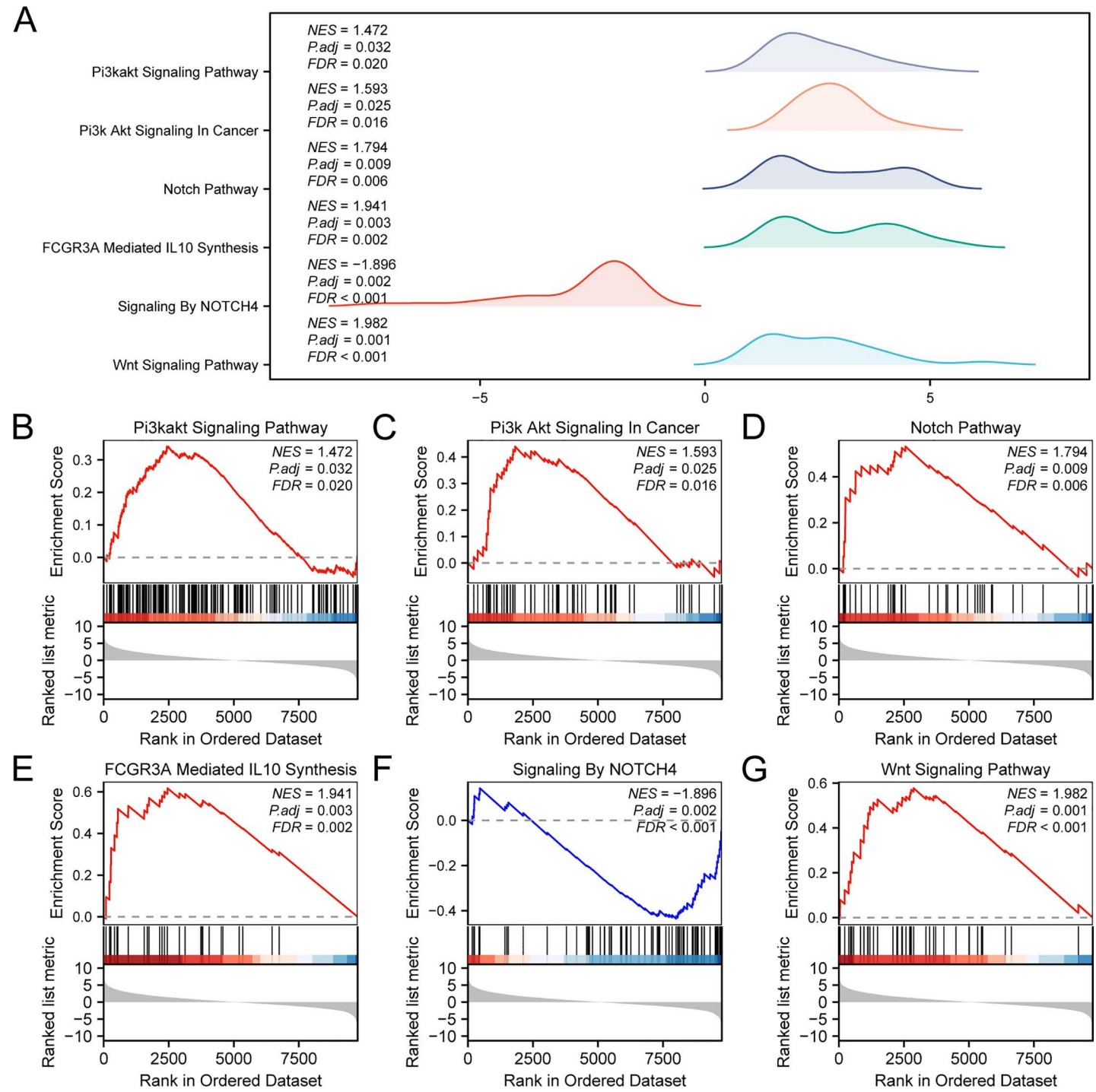

**Fig 4. GSEA for the combined GEO dataset. A.** Six biological function mountain maps of GSEA for the combined GEO dataset. B–G. Gene set enrichment analysis (GSEA) showed significant enrichment of genes within several key pathways, including the PI3K-Akt signaling pathway **(B)**, PI3K-Akt signaling in cancer **(C)**, Notch pathway **(D)**, FCGR3A-mediated IL10 synthesis **(E)**, signaling by NOTCH4 **(F)**, and Wnt signaling pathway **(G)**. t-MN, therapy-related myeloid neoplasm; GSEA, gene set enrichment analysis. The screening criteria for GSEA were adjusted $p < 0.05$ and FDR **(q)** < 0.25.

**Table 3. Results of GSEA for the combined GEO dataset.**

| ID | Setsize | Enrichment score | NES | p | Adjusted p | q | Rank |
|---|---|---|---|---|---|---|---|
| WP_WNT_SIGNALING_PATHWAY_AND_PLURIPOTENCY | 57 | 0.534689645 | 1.934459206 | 3.76E-05 | 0.000651435 | 0.000418147 | 2418 |
| WP_WNT_SIGNALING_PATHWAY | 41 | 0.577207023 | 1.981755869 | 8.49E-05 | 0.001250447 | 0.000802645 | 2873 |
| REACTOME_SIGNALING_BY_NOTCH4 | 70 | −0.435151113 | −1.896399841 | 0.000109244 | 0.001522774 | 0.000977447 | 1761 |
| REACTOME_FCGR3A_MEDIATED_IL10_SYNTHESIS | 27 | 0.617056218 | 1.940913114 | 0.000272987 | 0.003060717 | 0.001964631 | 2450 |
| PID_NOTCH_PATHWAY | 38 | 0.532032899 | 1.793863745 | 0.00127525 | 0.009497618 | 0.006096387 | 2557 |
| PID_IL6_7_PATHWAY | 41 | 0.501499654 | 1.721825695 | 0.002893399 | 0.016924643 | 0.010863689 | 3591 |
| REACTOME_PI3K_AKT_SIGNALING_IN_CANCER | 58 | 0.439680618 | 1.59274865 | 0.00503571 | 0.024569074 | 0.015770541 | 1810 |
| WP_PI3KAKT_SIGNALING_PATHWAY | 170 | 0.341229144 | 1.472280277 | 0.007202054 | 0.031530403 | 0.02023892 | 2478 |

GSEA, gene set enrichment analysis.

PI3K-Akt signaling pathway (Fig 4B), PI3K-Akt signaling in cancer (Fig 4C), Notch pathway (Fig 4D), FCGR3A-mediated IL10 synthesis (Fig 4E), signaling by NOTCH4 (Fig 4F), Wnt signaling pathway (Fig 4G), and various other biologically relevant functions and signaling pathways.

## Construction of the PPI network and screening of hub genes

Initially, we conducted a PPI analysis and constructed a network (Fig 5A) of 46 PRDEGs using the STRING database. Cytoscape software was used to visualize the PPI network, which revealed that 41 apoptosis-related PRDEGs were interconnected (Fig 5B). Subsequently, five algorithms from Cytoscape's cytoHubba plugin—namely, MCC, MNC, degree, EPC, and closeness, were applied to calculate and rank the scores of the 41 PRDEGs. The PPI network was then mapped by applying the top 10 genes identified by each of the five algorithms—namely, MCC (Fig 5C), MNC (Fig 5D), degree (Fig 5E), EPC (Fig 5F), and closeness (Fig 5G). The color gradient of the circle from red to yellow represents a score ranging from high to low. Finally, we identified the intersection of the genes from the five algorithms, which were analyzed and are illustrated in Fig 5H. The intersecting genes identified by the algorithm were designated as hub genes of the PRDEGs *Trp53*, *Mtor*, *Gpx3*, *Foxo3* and *Cybb*.

## Construction of the regulatory network

Initially, we retrieved the TF associated with hub genes from the ChIPBase database and constructed an mRNA-TF regulatory network, which was visualized using Cytoscape software (Fig 6A). This network included four hub genes and twenty-five transcription factors. For more information, please refer to S2 Table. Subsequently, we identified miRNAs associated with hub genes from the TarBase database and constructed an mRNA-miRNA regulatory network, which was visualized using the Cytoscape software (Fig 6B). This network comprised four hub genes and forty-five miRNAs. For more details, please refer to S3 Table.

## Differential expression verification and ROC curve analysis of hub genes

To investigate the expression differences of the hub genes in the combined GEO dataset, we utilized a group comparison figure (Fig 7A) to illustrate the expression levels of the five hub genes between the t-MN and control samples. The results indicated that the expression levels of the five hub genes, *Trp53*, *Mtor*, *Gpx3*, *Foxo3* and *Cybb*-were statistically significant ($p < 0.05$) in the t-MN samples compared to those in control samples from the integrated GEO dataset. Furthermore, ROC curves were plotted using the R package pROC based on the expression levels of hub genes from the integrated GEO dataset. The ROC curves (Fig 7B–7D) showed that the expression levels of *Trp53*, *Mtor*, *Gpx3*, *Foxo3* and *Cybb* exhibited high accuracy (AUC > 0.9) in classifying the t-MN and control samples (S4–S6 Tables).

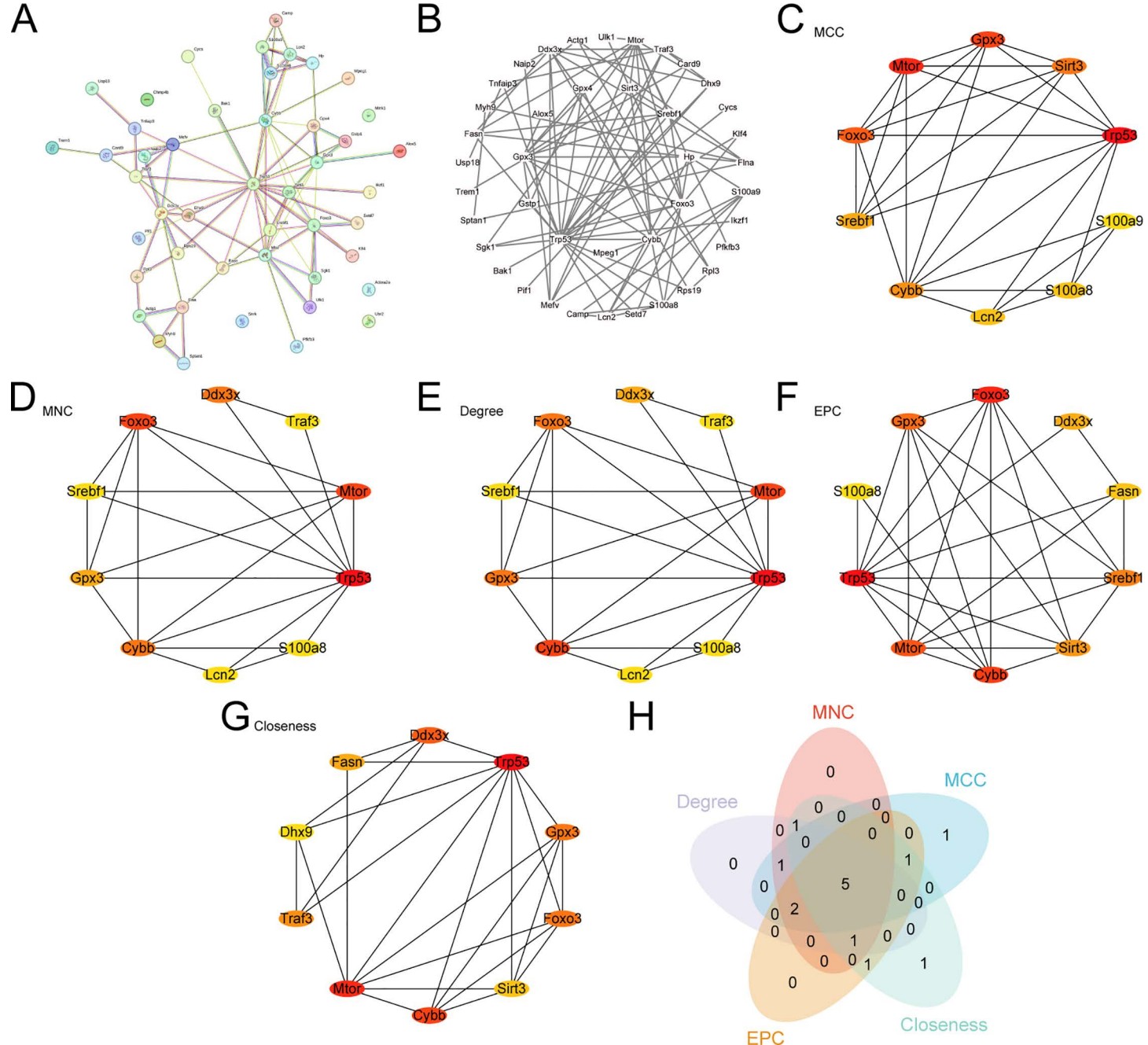

**Fig 5. PPI network and hub gene analysis.** A–B. PPI network of PRDEGs. STRING interaction network diagram **(A)**, Cytoscape interaction network diagram **(B)**. C–G. PPI network calculated by five algorithms of the cytoHubba plugin: MCC **(C)**, MNC **(D)**, degree **(E)**, EPC **(F)**, and closeness **(G)**. **H.** Venn diagram shows the intersection of the top 10 PRDEGs from the five algorithms of the cytoHubba plugin. PRDEGs, pyroptosis related differentially expressed genes; t-MN, therapy-related myeloid neoplasm; PPI, protein–protein interaction.

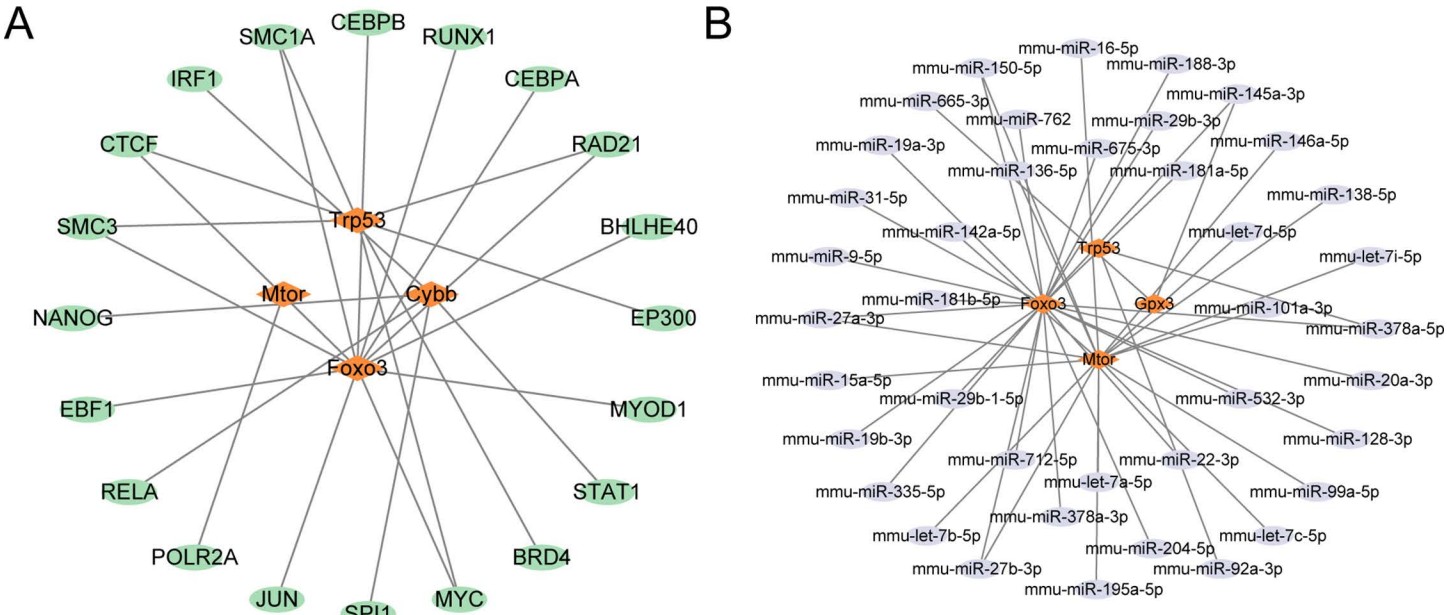

**Fig 6. Regulatory network of hub genes. A.** The mRNA-TF regulatory network of focal death-related differential genes (DEGs). B. mRNA-miRNA regulatory network of PRDEGs. TF, transcription factor. Orange represents the mRNA, green represents the TF, and purple represents the miRNA.

## Immunoinfiltration analysis for treatment-associated myeloid tumors (CIBERSORT)

The CIBERSORT algorithm was employed to calculate the abundance of 22 types of infiltrating immune cells in the combined GEO dataset. The immunoinfiltration analysis results were used to plot the proportion of immune cells in the combined GEO dataset (Fig 8A). The subgroup comparison graph (Fig 8B) illustrates the differences in immune cell infiltration abundance between the t-MN and control samples in the integrated GEO dataset. The analysis revealed statistically significant differences ($p < 0.05$) in the expression levels of four immune cell types between the t-MN and control samples: mast cells, neutrophils, NK cells, and plasma cells. Subsequently, the correlation results of the infiltration abundance of these four types of immune cells are shown by correlation heatmaps (Fig 8C). The results indicated that Mast Cells and NK Resting exhibited the highest positive correlation ($r = 0.64$), while Neutrophil Cells and NK Resting demonstrated the strongest negative correlation ($r = -0.90$). Finally, a correlation bubble map was used to illustrate the correlation between hub genes and the abundance of immune cell infiltration in the combined GEO dataset (Fig 8D). The correlation bubble map revealed a significant positive correlation between the hub gene *Cybb* and neutrophils ($r > 0.0$, $p < 0.05$), and a significant negative correlation between the hub gene *Cybb* and NK Resting immune cells ($r < 0.0$, $p < 0.05$).

## Discussion

Therapy-related myeloid neoplasms (t-MNs) represent a significant complication of cancer treatment, posing major threats to patient survival and quality of life [1]. Despite extensive research, the underlying mechanisms and key molecules involved in t-MN pathogenesis remain incompletely understood, creating a substantial knowledge gap [6]. While previous studies have focused on specific aspects of t-MNs, a comprehensive understanding of pyroptosis-related differentially expressed genes (PRDEGs) and their regulatory networks in t-MNs remains limited [51].

The present research—based on a publicly available, small-scale murine bone marrow RNA-seq dataset (GSE135866; n = 3 per group)—delves into t-MNs that gravely impact patient health and quality of life, particularly given the limitations of

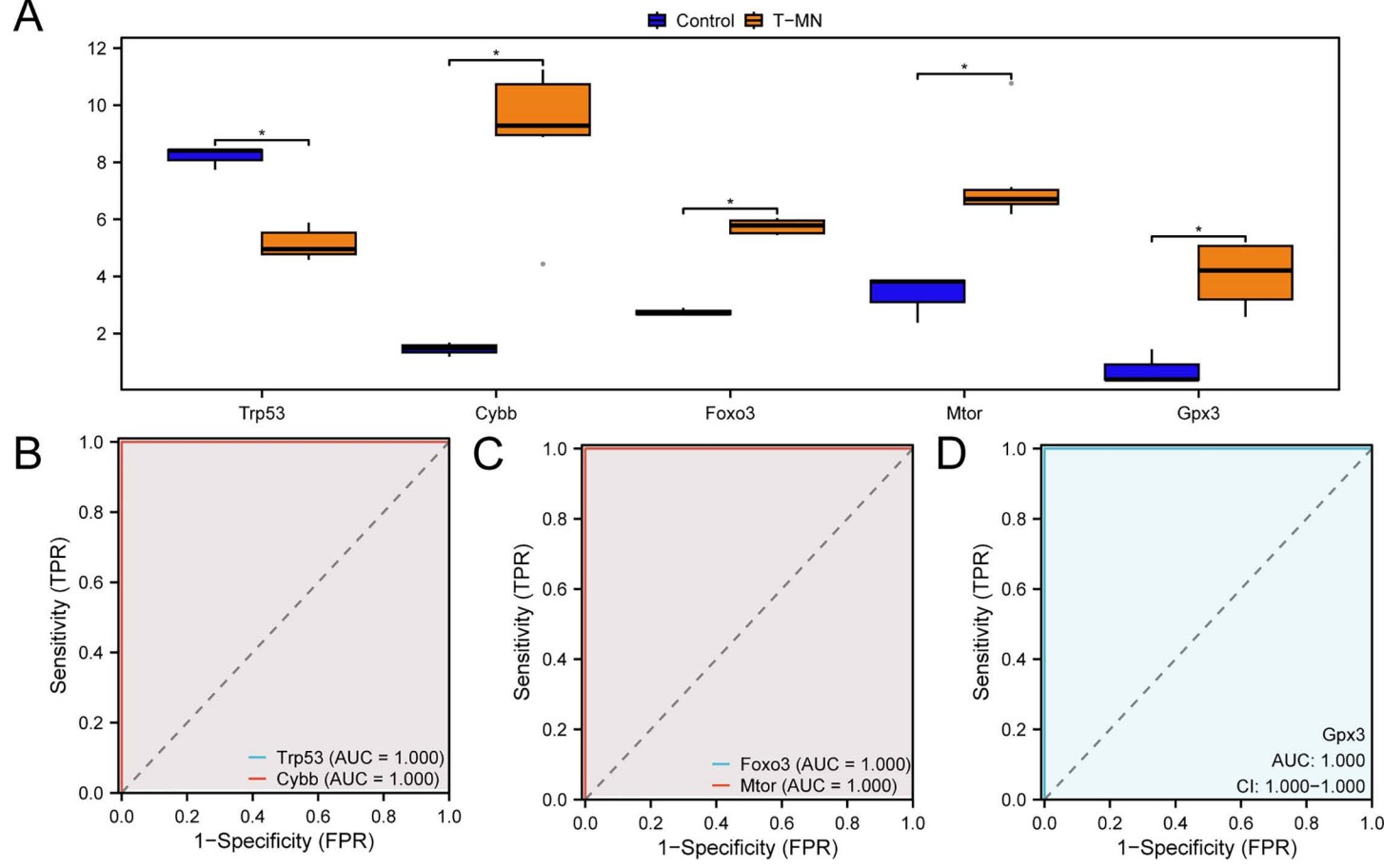

**Fig 7. Differential expression validation and ROC curve analysis. A.** Grouping comparison diagram of hub genes in the t-MN and control samples in the combined GEO dataset. B–D. ROC curve analysis of hub genes *Trp53, Cybb* **(B)**; *Foxo3, Mtor* **(C)**; and *Gpx3* (D) from the combined GEO dataset. * represents $p < 0.05$, which is statistically significant. t-MN, therapy-related myeloid neoplasm; ROC, receiver operating characteristic; AUC, area under the curve; TPR, true positive rate; FPR, false positive rate. Dark blue and orange denote the control and t-MN samples, respectively.

current therapeutic approaches. Given these constraints, our analysis is strictly exploratory and aims to depict the overall expression landscape and potential regulatory network of pyroptosis-related molecules in a preclinical t-MN model, rather than to support generalizable or causal conclusions. This underscores the urgent need for comprehensive research to identify new strategies for t-MN management. Through integrated bioinformatics analyses—including data acquisition, identification of PRDEGs, functional enrichment analyses (GO and KEGG), and networks construction—and acknowledging the limited statistical power due to the small sample size (n = 3 per group), we have generated preliminary insights into the molecular mechanisms underlying t-MNs. These findings should be interpreted at the hypothesis-generating level and serve as a reference for future studies with larger cohorts and independent models. Nevertheless, several pivotal findings emerged from our study. Most notably, we identified 46 PRDEGs that distinguish t-MN from control samples. These genes play crucial roles in various biological processes and offer significant insights into the molecular foundation of t-MN. Functional enrichment analysis has enhanced our understanding of their potential functions and implications. The identification and characterization of these PRDEGs not only augments the existing knowledge base but also suggest new avenues for therapeutic interventions and diagnostic markers for t-MN. Collectively, our findings—while limited by sample size and preclinical origin—provide a preliminary framework for future research aimed at improving patient care and outcomes. They

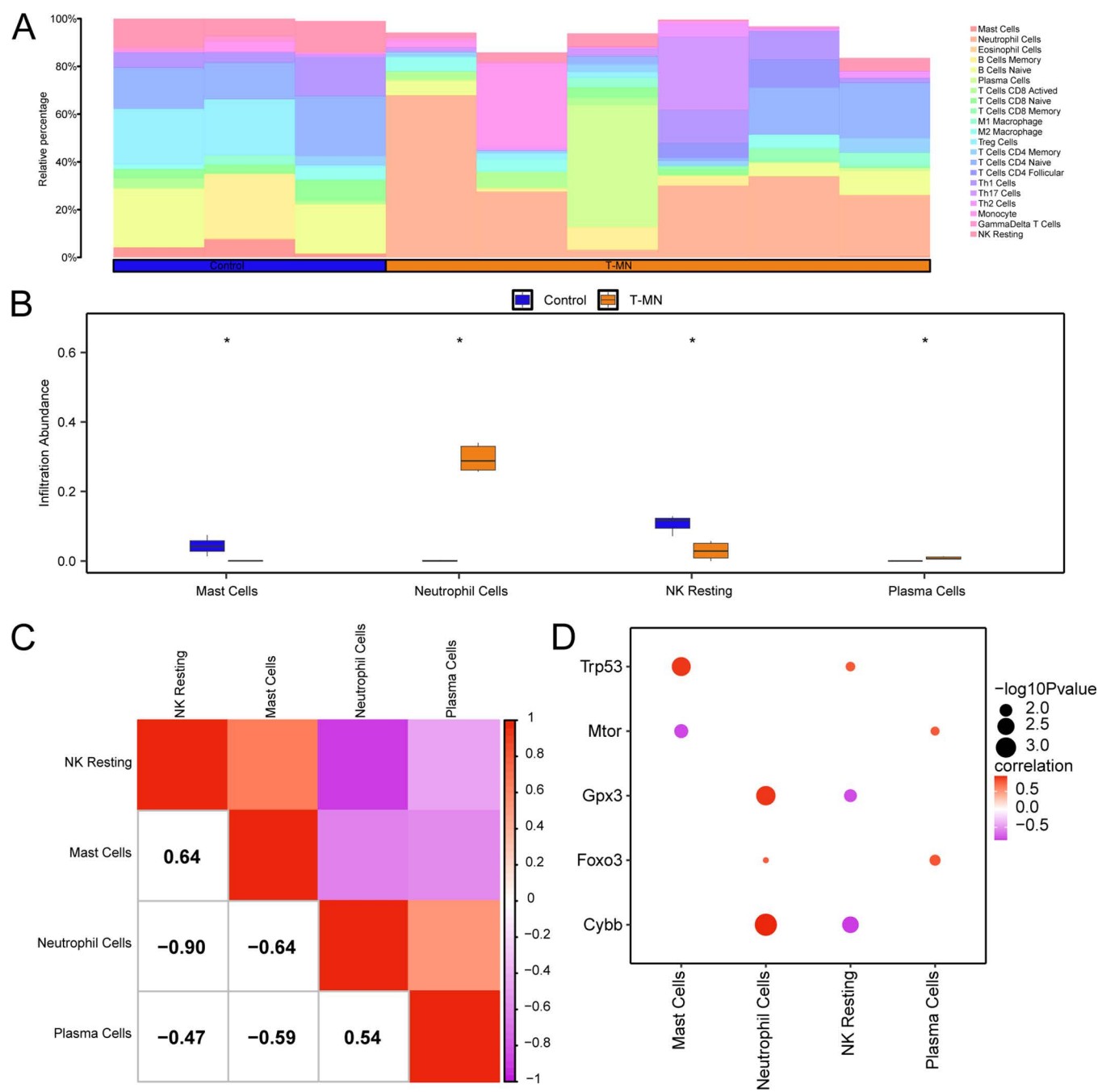

**Fig 8. Immune infiltration analysis using the CIBERSORT algorithm in the combined GEO dataset.** A–B. Bar chart (A) and group comparison chart (B) of the proportion of immune cells in the integrated GEO dataset (combined GEO dataset). **C.** Correlation heat map of the abundance of immune cell infiltration in the combined GEO dataset. **D.** Bubble map of correlation between hub genes and the abundance of immune cell infiltration in the combined GEO dataset. t-MN, therapy-related myeloid neoplasm. * represents $p < 0.05$, which has statistical significance. The absolute value of the correlation coefficient ($r$) is categorized as follows: a value below 0.3 indicates weak or no correlation, values between 0.3 and 0.5 signify weak correlation, values between 0.5 and 0.8 represent moderate correlation, and values above 0.8 indicate strong correlation. The t-MN samples are shown in orange, whereas the control samples are denoted in dark blue. In correlation visualizations, purple indicates negative correlation, and red signifies positive correlation, with the intensity of the color indicating the strength of the correlation.

should be viewed as a starting point for hypothesis-driven validation, not as definitive evidence. This study offers an initial, data-driven characterization of pyroptosis-related gene expression in a murine t-MN model and may inform, rather than substantially advance, our understanding of t-MNs. Its promise lies in guiding—not concluding—future efforts to address this challenging condition.

The identification of 46 PRDEGs, such as *Bak1* [52], *Pif1* [53], *Sptan1*, and others, which exhibited significant differential expression in t-MN samples, offers valuable genetic insights. An important consideration is whether these genes play similar roles in other cancer types, as many may exhibit pleiotropic effects and comparable functions across different malignancies [54,55]. Understanding the involvement of these genes in pyroptosis and their influence on tumor progression is equally essential. Pyroptosis is a highly regulated form of programmed cell death that significantly affects tumor growth and metastasis [56,57]. Genes like *Pif1* might be involved in DNA damage repair or chromatin remodeling, directly affecting genomic stability and tumor development [58]. Elucidating the molecular mechanisms by which these genes regulate pyroptosis and interact with other cellular processes will enhance our understanding of the complex networks underlying t-MNs and other cancers. Furthermore, the potential of these genes as biomarkers for clinical diagnosis and prognostic assessment warrants careful consideration. The distinct expression patterns in t-MN samples suggest utility as reliable indicators for disease detection and monitoring. Longitudinal studies tracking gene expression over time could provide insights into treatment responses and disease recurrence. However, further validation and extensive clinical studies are required to confirm the clinical utility and reliability of these potential biomarkers. In summary, investigation of these genes in t-MNs not only provides insights into the molecular underpinnings of the disease but also holds promise for translational applications in cancer diagnosis and therapy.

Although genes such as *Trp53*, *Mtor*, and *Foxo3* are well-established players in myeloid malignancies, our study provides a novel perspective by delineating their specific roles in the pyroptosis pathway within t-MNs. Traditionally, *Trp53* mutations are recognized as critical biomarkers in t-MNs, associated with genomic instability and poor prognosis [12,22]. However, beyond its canonical functions in apoptosis and DNA repair, emerging evidence indicates that p53 (encoded by *Trp53*) actively regulates pyroptosis. For instance, p53 suppresses tumor growth by prompting pyroptosis in non-small-cell lung cancer [59]. In the context of t-MNs, where DNA damage is a key initiating factor, this p53-mediated pyroptotic pathway may represent a double-edged sword: it could eliminate damaged cells but also contribute to inflammatory microenvironment alterations that favor leukemogenesis. This pyroptosis-focused mechanism distinguishes our findings from conventional *Trp53* mutation studies, which primarily emphasize genomic instability, and highlights the potential for targeting p53-dependent pyroptosis as a therapeutic strategy in t-MNs. Similarly, *Mtor*, a central regulator of cellular metabolism and growth, has been implicated in myeloid neoplasms through its role in promoting cell proliferation. Our enrichment analysis revealed involvement in the AMPK signaling pathway, which negatively regulates mTOR. Recent studies suggest that mTOR inhibition can induce pyroptosis via dysregulation of autophagy and inflammasome activation [60,61]. In t-MNs, chemotherapy-induced metabolic stress may activate AMPK, leading to mTOR suppression and subsequent pyroptosis, which could influence clonal selection and disease progression. This metabolic-pyroptotic axis offers a new dimension to *Mtor*'s function beyond its known proliferative effects. Furthermore, FoxO3, a transcription factor involved in oxidative stress response, is often associated with apoptosis in leukemia. Our data indicate its connection to pyroptosis through antioxidant activity and inflammatory regulation. FoxO3 can modulate the expression of pyroptosis-related genes like *NLRP3* under oxidative stress conditions [62], suggesting a cross-talk between oxidative damage and pyroptotic cell death in t-MNs. Unlike conventional biomarkers that focus on FoxO3's role in cell cycle arrest, our findings emphasize its involvement in inflammatory cell death pathways, providing a unique angle for targeting the redox-pyroptosis interplay in t-MNs. In comparison to existing biomarkers such as *Trp53* mutation status, which serves as a static indicator of genomic damage, our pyroptosis-related hub genes dynamic regulation of cell death and inflammation offers a more functional insight into t-MN pathogenesis. While *Trp53* mutations are prognostic, they do not fully capture the complex cell death mechanisms altered by therapy. Our approach identifies actionable pathways (e.g., AMPK-mTOR-pyroptosis) that

could be exploited for combination therapies, enhancing the specificity of interventions. Thus, labeling these genes as "pyroptosis-related" not only underscores their pleiotropic roles but also reveals novel mechanistic links that may lead to innovative biomarkers and targeted treatments for t-MNs.

The identification of *Trp53* and *Mtor* as central hub genes underscores the complex, pleiotropic nature of regulatory networks in t-MN pathogenesis, particularly as they relate to pyroptosis. As master regulators of cellular stress responses, their involvement suggests a critical interface between pyroptotic cell death and other fundamental processes like genomic integrity and metabolic signaling. The tumor suppressor p53, encoded by *Trp53*, is a well-known regulator of apoptosis and ferroptosis, but its role in pyroptosis is increasingly recognized. p53 can transcriptionally activate key components of the pyroptotic pathway, such as genes for gasdermin family members (*GSDME*) and NLRP3 inflammasome components, thereby sensitizing cells to pyroptosis upon cytotoxic stress—a common feature of the chemotherapeutic agents that predispose to t-MN development [63–65]. Conversely, mutant p53 may acquire gain-of-function properties that suppress pyroptosis, promoting cell survival and clonal expansion. Similarly, mTOR (mechanistic target of rapamycin) serves as a central integrator of nutrient and energy status, cell growth, and autophagy. The AMPK signaling pathway, identified as enriched in our KEGG analysis, is a primary upstream inhibitor of mTOR. AMPK activation and subsequent mTOR inhibition can promote autophagy, which has a dual and context-dependent relationship with pyroptosis [63]. While autophagy can dampen pyroptosis by clearing damaged organelles like mitochondria (thus reducing ROS and inflammasome activation), it can also facilitate the presentation of pyroptotic stimuli. Furthermore, mTOR signaling can directly influence the expression and activity of inflammasome components. Therefore, the dysregulation of *Mtor* observed in our t-MN model likely contributes to an altered pyroptotic threshold, potentially enabling the survival of damaged pre-leukemic cells. The interplay between these pleiotropic genes creates a sophisticated regulatory circuit where DNA damage, metabolic stress, and inflammatory cell death converge, potentially offering novel nodes for therapeutic intervention aimed at selectively inducing pyroptosis in t-MN clones.

Using GO and KEGG pathway enrichment analyses and GSEA, we found that PRDEGs were notably enriched in pathways related to autophagy, inflammatory response regulation, and the NOD-like receptor signaling pathway. This finding suggests that autophagy and inflammatory response regulation may be key pathogenic mechanisms in t-MNs and may play a crucial role in its progression. Autophagy, a complex regulator of cellular homeostasis, intersects with inflammatory processes [66]. In t-MNs, autophagy facilitates the removal of damaged organelles and proteins, thereby influencing disease progression. Conversely, inflammatory responses triggered by diverse factors can modulate autophagy, as exemplified by the cytokines released during inflammation that alter autophagy pathways. Key mediators, such as *Trp53* and *Cybb* play dual roles in this intricate interplay. Comparative analysis of signaling pathway activities between normal and tumor tissues often reveals stark contrasts. While normal tissues exhibit stringent regulation to maintain equilibrium, genetic and signaling anomalies provoke irregular activation or suppression of pathways in t-MNs. The identified hub genes likely contribute to these discrepancies. Elucidating these variations is crucial for developing therapies that selectively correct dysregulated pathways in tumors, while preserving normal tissue integrity. Further research is essential to fully understand these differences and their therapeutic implications.

Intricate tapestry interactions occur within the realm of immune infiltration. Initially, diverse immune cell populations engaged in sophisticated dance within the tumor microenvironment. For instance, mast cells unleash mediators that modulate the immune response and potentially exert both pro- and anti-tumor effects. Neutrophils also exhibit multifaceted roles, occasionally fostering tumor expansion or engaging in antitumor defenses. In their resting state, NK cells are pivotal for innate immune surveillance by directly targeting tumor cells. Plasma cells contribute to the immunological attack of the body against tumors through antibody production. These cellular factors may collaborate harmoniously or counteract each other. For instance, mast cells orchestrate the mobilization or activation of neutrophils, whereas NK cells collaborate with plasma cells to amplify immune effector mechanisms. Such concerted or conflicting behaviors significantly shape the tumor microenvironment and its evolution. Determining the connection between pivotal genes and specific immune cells

provides an opportunity to predict therapeutic outcomes. If certain key genes strongly correlate with distinct patterns of immune cell infiltration, this insight could illuminate a tumor's potential response to immunotherapy. For example, a scenario in which a particular gene associated with heightened mast cell infiltration corresponds to an improved response to a given immunotherapy could emerge as a critical biomarker for personalized treatment strategies. Unraveling these complex interrelationships will pave the way for personalized medicine, enabling the identification of patients who are most likely to benefit from targeted immunotherapeutic interventions. Multiple strategies can be explored to enhance the current treatment plans by leveraging immune infiltration characteristics. One effective approach is to integrate immunotherapies with agents that target or modulate the function of specific immune cell subsets. For example, if neutrophils are identified as detrimental to a specific type of tumor, targeting them in conjunction with immunotherapy could potentially improve treatment efficacy. Additionally, real-time monitoring of immune infiltration patterns during therapy can facilitate timely adjustment of treatment plans. In cases where a patient exhibits no response or changes in immune cell composition, alternative or combination therapies may be considered. Furthermore, the development of novel therapeutic agents that specifically target the interactions between immune cells and hub genes or that favorably manipulate the immune microenvironment holds significant promise for improving treatment outcomes. In summary, a comprehensive understanding of the intricate interactions between different immune cell types, the relationships between hub genes and immune cells, and the application of this knowledge to optimize treatment plans is essential in the field of cancer immunotherapy. Further research is necessary to fully utilize these aspects and translate them into effective clinical strategies for enhanced patient care.

The exceptional classification accuracy (AUC > 0.9) of the hub genes *Trp53*, *Cybb*, *Foxo3*, *Mtor*, and *Gpx3* in distinguishing t-MN patients from controls represents a significant diagnostic advance. However, translating these findings into clinical utility requires rigorous validation and further exploration. First, the classification performance of these genes must be validated in independent cohorts to confirm generalizability and reliability, mitigating risks of overfitting and population-specific bias inherent to single-cohort studies. Second, exploring combinations of these hub genes with other molecular markers—such as additional genetic alterations, epigenetic modifications, or protein biomarkers—could yield a more comprehensive and accurate diagnostic model, potentially enabling earlier and more precise detection of t-MN. Additionally, it is important to assess whether the diagnostic performance of these genes remains consistent across different t-MN stages or subtypes. Such insights could inform personalized strategies, as certain genes may be more relevant in early-stage disease while others are pivotal in advanced subtypes. In summary, while the high classification accuracy of these genes is notable, future work must focus on independent validation, development of combined biomarker panels, and evaluation across disease stages to fully realize their diagnostic and therapeutic potential.

Although this study advances our understanding of the pyroptosis-related mechanisms in t-MNs, it is essential to emphasize that this research is based on mouse bone marrow data, representing a preclinical exploratory analysis. Several limitations must be acknowledged. The relatively small sample size (n = 9) may limit the statistical power and generalizability of our findings, potentially increasing the risk of overfitting and reducing the robustness of the biomarker signatures identified. Reliance on retrospective public datasets introduces potential batch effects and clinical metadata gaps, particularly regarding treatment-dose correlations and ancestral diversity. Additionally, the absence of wet-lab validation leaves mechanistic predictions for key genes (e.g., *Cybb* and *NLRP3*) and pathway enrichment (AMPK/PI3K-Akt signaling) that were not confirmed in experimental models. Although our machine learning classifier demonstrated diagnostic potential (AUC > 0.9), its clinical translation faces practical barriers, including unresolved cost-effectiveness ($1,500/sample RNA-seq vs. conventional cytogenetics), unproven performance in liquid biopsies, and a lack of CLIA-certified assay platforms. The static bulk sequencing approach fails to capture the temporal dynamics of clonal evolution or distinguish between preleukemic and transformation-phase molecular events, compounded by sample size limitations in detecting rare (< 5% prevalence) driver mutations. Given the exploratory nature of this study, our conclusions should be interpreted as hypothesis-generating rather than definitive. Future studies with expanded cohort sizes and experimental validations

are warranted to verify the reproducibility and clinical relevance of these findings. Furthermore, therapeutic strategies predicated on pyroptosis induction risk unintended consequences, as primate models reveal suboptimal bone marrow-targeting efficiency (< 5%) and adaptive upregulation of anti-pyroptotic factors (e.g., *BCL2L1*) in 40% of patient-derived xenograft (PDX) models. Moreover, *Cybb* inhibition may exacerbate susceptibility to infection through NADPH oxidase suppression. These constraints underscore the need for multicenter prospective cohort studies, functional validation using t-MN-specific cellular models, and phase 0/1 trials incorporating spatial and dynamic biomarker monitoring to bridge computational insights and clinical implementation.

In conclusion, this exploratory bioinformatic analysis—based on a public, small-scale mouse bone marrow RNA-seq dataset—identified key genes and signaling pathways involved in t-MNs, revealing 46 PRDEGs. Comprehensive enrichment analysis predicted that PRDEGs are involved in autophagy, inflammatory response regulation, apoptosis, NOD-like receptor signaling, AMPK signaling, PI3K-Akt signaling, and Notch signaling. Immunoinfiltration analysis showed a significant positive correlation between the hub gene *Cybb* and neutrophils ($r > 0.0$, $p < 0.05$), and a significant negative correlation with resting NKg cells ($r < 0.0$, $p < 0.05$). These results reflect the global expression pattern and putative regulatory associations of pyroptosis-related molecules in a preclinical setting and are intended solely to generate hypotheses for follow-up studies with larger sample sizes and human validation. These findings enhance our preliminary understanding of the molecular mechanisms underlying this disease at the descriptive and correlative level, and suggest new avenues for diagnostic and therapeutic strategies. They do not establish mechanistic causality or clinical applicability. Moving forward, efforts should focus on rigorous experimental validation in independent models and eventual clinical correlation to determine whether these hypothesis-generating observations can contribute to precision medicine.

## Supporting information

**S1 Table List of 455 PRGs.**
(CSV)

**S2 Table List of four hub genes and twenty-five transcription factors.**
(CSV)

**S3 Table List of four hub genes and forty-five miRNAs.**
(CSV)

**S4 Table 95% CIs for ROC curves of Figure 7.**
(CSV)

**S5 Table DeLong test p-values for Trp53 and Cybb in Figure 7B.**
(CSV)

**S6 Table DeLong test p-values for Foxo3 and Mtor in Figure 7C.**
(CSV)

## Author contributions

**Conceptualization:** Jing Cheng, Xiaohui Zhu.

**Data curation:** Jing Cheng, Weiyue Fang.

**Formal analysis:** Jing Cheng, Weiyue Fang.

**Funding acquisition:** Xiaoxia Zhan.

**Investigation:** Jing Cheng, Hongxia Tan.

**Methodology:** Jing Cheng, Weiyue Fang, Hongxia Tan, Xiaoxia Zhan.

**Project administration:** Xiaoxia Zhan, Xiaohui Zhu.

**Software:** Weiyue Fang.

**Supervision:** Xiaoxia Zhan, Xiaohui Zhu.

**Writing – original draft:** Jing Cheng, Weiyue Fang, Hongxia Tan.

**Writing – review & editing:** Xiaoxia Zhan, Xiaohui Zhu.

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
