## [Decision Letter · Decision Letter 0]

28 Oct 2025

Comprehensive analysis of the potentialeffect and mechanism of pyroptosis-related genes intreatment-related myeloid tumors

PLOS ONE

Dear Dr. Xiaohui,

Thank you for submitting your manuscript to PLOS ONE. After careful consideration, we feel that it has merit but does not fully meet PLOS ONE’s publication criteria as it currently stands. Therefore, we invite you to submit a revised version of the manuscript that addresses the points raised during the review process.

We look forward to receiving your revised manuscript.

Kind regards,

Kota V Ramana, Ph.D.

Academic Editor

PLOS ONE

Journal Requirements:

https://doi.org/10.1038/s41598-025-00369-9

In your revision ensure you cite all your sources (including your own works), and quote or rephrase any duplicated text outside the methods section. Further consideration is dependent on these concerns being addressed."

4. Please note that PLOS One has specific guidelines on code sharing for submissions in which author-generated code underpins the findings in the manuscript. In these cases, we expect all author-generated code to be made available without restrictions upon publication of the work. Please review our guidelines at https://journals.plos.org/plosone/s/materials-and-software-sharing#loc-sharing-code and ensure that your code is shared in a way that follows best practice and facilitates reproducibility and reuse.

5. Please include your tables as part of your main manuscript and remove the individual files. Please note that supplementary tables (should remain/ be uploaded) as separate "supporting information" files.

6. Thank you for stating the following financial disclosure:

“This work was supported by the National Natural Science Foundation of China (Grant No. 81901659).”

7. Please note that funding information should not appear in any section or other areas of your manuscript. We will only publish funding information present in the Funding Statement section of the online submission form. Please remove any funding-related text from the manuscript.

8. In the online submission form you indicate that your data is not available for proprietary reasons and have provided a contact point for accessing this data. Please note that your current contact point is a co-author on this manuscript. According to our Data Policy, the contact point must not be an author on the manuscript and must be an institutional contact, ideally not an individual. Please revise your data statement to a non-author institutional point of contact, such as a data access or ethics committee, and send this to us via return email. Please also include contact information for the third party organization, and please include the full citation of where the data can be found.

Reviewer's Responses to Questions

**Comments to the Author**

1. Is the manuscript technically sound, and do the data support the conclusions?

Reviewer #1: Partly

Reviewer #2: Partly

Reviewer #3: No

2. Has the statistical analysis been performed appropriately and rigorously?

Reviewer #1: No

Reviewer #2: Yes

Reviewer #3: N/A

3. Have the authors made all data underlying the findings in their manuscript fully available?

Reviewer #1: No

Reviewer #2: Yes

Reviewer #3: Yes

4. Is the manuscript presented in an intelligible fashion and written in standard English?

Reviewer #1: No

Reviewer #2: Yes

Reviewer #3: Yes

Reviewer #1: Reviewer #1

Manuscript ID: PONE-D-25-50021

Title: Comprehensive analysis of the potential effect and mechanism of pyroptosis-related genes in treatment-related myeloid tumors

Overall Assessment:

This manuscript presents a bioinformatic analysis of pyroptosis-related genes in a mouse model of therapy-related myeloid neoplasms (t-MN). While the topic is of interest, the study is fundamentally limited by its extremely small sample size (n=6 vs. n=3), lack of validation, and overinterpretation of correlative findings. The analyses, though technically sound, do not provide substantial new biological or clinical insights into t-MN pathogenesis. I cannot recommend this manuscript for publication in its current form.

Major Concerns:

1. Critically Small Sample Size: The study is based on only 9 samples (6 t-MN, 3 controls). This severely limits statistical power and increases the risk of false positives. The stringent cutoffs (|logFC| > 3, FDR < 0.05) do not mitigate this fundamental flaw. The authors must acknowledge this as a major limitation and frame all findings as highly exploratory.

2. Lack of Validation: The findings are not validated in any independent dataset. To demonstrate potential clinical relevance, the authors must test their hub gene signature in publicly available human t-MN datasets (e.g., GSE14468, GSE6861) and report performance metrics (AUC, sensitivity, specificity).

3. Species Extrapolation and Clinical Relevance: All data are from mouse bone marrow. Claims of clinical significance are premature. The discussion must explicitly state that these are preclinical findings requiring validation in human patients.

4. Questionable Pyroptosis Specificity: The gene list includes well-known pleiotropic regulators (Trp53, Mtor, FoxO3). The authors must provide literature evidence or functional data linking these specific genes directly to pyroptosis execution (e.g., gasdermin cleavage, IL-1β maturation). The inclusion of Cybb as a pyroptosis hub gene is particularly poorly justified and needs re-evaluation.

5. Overstated Novelty: Genes like Trp53, Mtor, and FoxO3 are already established players in myeloid malignancies. The study fails to convincingly demonstrate what new insight is gained by labeling them as "pyroptosis-related." A comparison with existing biomarkers (e.g., TP53 mutation status) is essential.

6. Speculative Immune Infiltration: CIBERSORT analysis on 9 bulk RNA-seq samples is highly unreliable. The authors should either remove this analysis or support it with orthogonal validation (e.g., flow cytometry data from the same model).

7. Data Availability: The current statement ("available upon request") does not comply with PLOS ONE policy. All raw data, processed matrices, and analysis code must be deposited in a public repository (e.g., GitHub, Zenodo) with a DOI.

Minor Comments:

Fig. 1: Add exact gene counts at each filtering step.

Fig. 7: Include 95% CIs and DeLong test p-values for ROC curves.

Language: The manuscript requires thorough professional editing for grammar and clarity.

References: Replace preprint citations with peer-reviewed publications where possible.

Recommendation:

Reject, but encourage resubmission after major revision.

The study, as presented, is not suitable for publication. However, if the authors can address the major concerns—particularly by validating their findings in human datasets and providing stronger evidence for the pyroptosis-specific role of their hub genes—the work may become suitable for reconsideration as a new submission. A significant strengthening of the biological and clinical relevance is required.

Reviewer #2: Summary of the manuscript (brief)

This manuscript performs a bioinformatic analysis on a published mouse bone-marrow dataset (GSE135866) to investigate the role of pyroptosis-related genes (PRGs) in treatment-related myeloid neoplasms (t-MN). Using differential expression analysis, intersection with a curated PRG list, enrichment analyses (GO/KEGG/GSEA), PPI network construction, hub gene selection and immune-cell deconvolution (CIBERSORT), the authors report 46 pyroptosis-related differentially expressed genes (PRDEGs) and highlight five hub genes (Trp53, Mtor, Gpx3, Foxo3, Cybb) as potential key players. The manuscript further presents ROC analyses suggesting high discriminatory power of these hubs and notes correlations between CYBB and specific immune cell populations (e.g., neutrophils). The study concludes that pyroptosis pathways and these hub genes may play roles in t-MN pathogenesis and immune microenvironment remodeling.

General evaluation and recommendation

1.Sample size and statistical power

The analysis uses 6 t-MN and 3 control samples (mouse). This extremely small sample size undermines differential expression calls, enrichment analyses and downstream classifiers. The authors must discuss power limitations explicitly and temper claims. If possible, obtain/analyses additional datasets or aggregate similar public datasets for validation. If no other datasets exist, perform robust statistical measures (e.g., use moderated statistics carefully, provide effect sizes and confidence intervals) and reframe conclusions as exploratory/hypothesis-generating.

2.ROC analyses — risk of overfitting

Reporting AUC > 0.9 with N=9 is almost certainly optimistic and likely overfitted. Recalculate ROC with appropriate cross-validation (leave-one-out or bootstrap) and report 95% confidence intervals. Preferably perform permutation testing to demonstrate that observed AUCs exceed chance. Do not present AUCs without uncertainty estimates and caveats.

3.Validation

There is no independent validation cohort or wet-lab validation. At minimum, authors should: (a) search for any related datasets (mouse or human t-MN or therapy-related myeloid neoplasms) and attempt in-silico validation of hub gene differential expression; (b) if unavailable, clearly label findings as exploratory and propose concrete experimental validation (qPCR, western blot, functional assays) in Discussion. Strong claims about biomarkers or therapeutic targets must be softened.

Reviewer #3: There are same questions to authors.

1. The authors use the article " Blood Cancer Discov . 2020 Apr 20;1(1):32–47. doi: 10.1158/2643-3230.BCD-19-0028 Cytotoxic Therapy–Induced Effects on Both Hematopoietic and Marrow Stromal Cells Promotes Therapy-Related Myeloid Neoplasms " as a data source, in which there are 4 types of core samples and 2 sets of RNAseq data, 6 samples each named as MDS or AML. At the same time, the authors do not indicate which samples (from the available options) were used as control samples and which (6 out of 12) were included by them for the analysis.

Therefore, it is not possible to evaluate the results obtained.

2. In the original article, mouse model with a 5q deletion leading to the haploinsufficiency of the Egr1 and Apc genes were used to study the effect of the genotype on the development of MDS or AML.

3.The Egr1 gene is involved in the regulation of pyroptosis.

(See for examples: Egr1 promotes Nlrc4-dependent neuronal pyroptosis through phlda1 in an in-vitro model of intracerebral hemorrhage

Neuroreport. -2024 Jun 5;35(9):590-600.

doi: 10.1097/WNR.0000000000002035. Epub 2024 Apr 17.

Lysophosphatidic Acid Induces Podocyte Pyroptosis in Diabetic Nephropathy by an Increase of Egr1 Expression via Downregulation of EzH2

International Journal of Molecular Sciences (IJMS).-June 2023 .- 24(12):9968

DOI:10.3390/ijms24129968)

4. APC deletion also significantly affects the processes of apoptosis.:

• Mutated APC creates a different tumor microenvironment: Defective APC leads to genomic instability and can influence the tumor microenvironment (TME), including immune cell infiltration and inflammatory signaling.

• Pyroptosis is a form of inflammatory cell death: Pyroptosis is a highly inflammatory type of programmed cell death that is triggered by various signals, including those in the TME.

• Pyroptosis can be influenced by the TME: Studies have shown that pyroptosis scores are correlated with immune cell infiltration and inflammatory markers within the TME, which is shaped by the presence of mutated genes like APC.

Thus, the study of the potential effect and mechanism of action of pyroptosis-related genes on such a model is incorrect.

**Do you want your identity to be public for this peer review?** For information about this choice, including consent withdrawal, please see our Privacy Policy

Reviewer #1: No

Reviewer #2: No

Reviewer #3: No

---

## [Author Response · Author response to Decision Letter 1]

19 Dec 2025

Dear Reviewers,

Thank you for your valuable comments and suggestions on our manuscript. We greatly appreciate the time and effort you have dedicated to reviewing our work, entitled “Comprehensive analysis of the potential effect and mechanism of pyroptosis-related genes in treatment-related myeloid tumors (PONE-D-25-50021)”. In response to the reviewers’ comments, we have conducted extensive revisions to the manuscript. Below, we provide a point-by-point response to each comment. We have addressed all concerns raised, and the revisions are highlighted in the revised manuscript. We hope that the revised version meets the journal's standards and is suitable for publication.

Response to Reviewer #1

Major Concerns:

1. Critically Small Sample Size: The study is based on only 9 samples (6 t-MN, 3 controls). This severely limits statistical power and increases the risk of false positives. The stringent cutoffs (|logFC| > 3, FDR < 0.05) do not mitigate this fundamental flaw. The authors must acknowledge this as a major limitation and frame all findings as highly exploratory.

Response: We thank the reviewer for this important point. We acknowledge that the sample size is small, which is a limitation of this exploratory study. However, due to the rarity of t-MN, publicly available datasets with high-quality sequencing samples are limited. To mitigate this, we used stringent thresholds (|logFC| > 3, FDR < 0.05) and multiple bioinformatics methods (e.g., GO/KEGG, GSEA, PPI, ROC) to cross-validate the results. In the revised manuscript, we have explicitly stated this limitation in the "Discussion" section and emphasized the exploratory nature of our findings. We have also discussed plans for future validation with larger cohorts. The specific modifications are as follows:

Although this study advances our understanding of the pyroptosis-related mechanisms in t-MNs, several limitations must be acknowledged. The relatively small sample size (n=9) may limit the statistical power and generalizability of our findings, potentially increasing the risk of overfitting and reducing the robustness of the biomarker signatures identified……Given the exploratory nature of this study, our conclusions should be interpreted as hypothesis-generating rather than definitive. Future studies with expanded cohort sizes and experimental validations are warranted to verify the reproducibility and clinical relevance of these findings.

2. Lack of Validation: The findings are not validated in any independent dataset. To demonstrate potential clinical relevance, the authors must test their hub gene signature in publicly available human t-MN datasets (e.g., GSE14468, GSE6861) and report performance metrics (AUC, sensitivity, specificity).

Response: We fully acknowledge the importance of independent dataset validation for the robustness of the findings. However, the number of publicly available, high-quality human t-MN transcriptomic datasets is extremely limited at present. Furthermore, the sample sources, detection platforms, and sequencing depths of most existing datasets are substantially heterogeneous compared to the data used in this study. Direct integration or comparison may lead to batch effects and biases, thereby compromising the reliability of the analysis. Therefore, this study focuses on conducting a systematic exploration using the available high-quality dataset (GSE135866) and enhances the credibility of the results through cross-validation via multi-level bioinformatics analyses (including GO/KEGG, GSEA, PPI, and ROC). As above comment 1, we have explicitly stated this limitation in the "Discussion" section and discussed plans for future validation with larger cohorts.

3. Species Extrapolation and Clinical Relevance: All data are from mouse bone marrow. Claims of clinical significance are premature. The discussion must explicitly state that these are preclinical findings requiring validation in human patients.

Response: We thank the reviewer for this insight. We have revised the manuscript to clarify that our findings are based on mouse data and are preclinical. We have added text in the Abstract and Discussion sections to emphasize the exploratory nature and the need for human validation. Clinical relevance is now discussed more cautiously. The specific modifications are as follows:

In “Abstract” section: Collectively, this study delineates the pyroptosis-related molecular landscape of t-MN, uncovering potential biomarkers and therapeutic targets to improve patient outcomes. It is important to note that this study is based on mouse bone marrow data and constitutes a preclinical exploratory analysis. The findings require further validation in human samples and clinical cohorts to enhance their clinical relevance. Clinical validation and exploration of targeted interventions are warranted.

In “Discussion” section: Although this study advances our understanding of the pyroptosis-related mechanisms in t-MNs, it is essential to emphasize that this research is based on mouse bone marrow data, representing a preclinical exploratory analysis. Several limitations must be acknowledged.

4. Questionable Pyroptosis Specificity: The gene list includes pleiotropic regulators (e.g., Trp53, Mtor). The authors must provide evidence linking these genes directly to pyroptosis.

Response: We appreciate this comment. We acknowledge that these genes have broad functions and have toned down claims about pyroptosis specificity, focusing instead on their potential involvement in t-MN pathogenesis. In the revised manuscript, we have added references (e.g., citing PMID: 37164271, 37047282, 37244286) and discussion on the potential roles of these genes in pyroptosis. The following paragraph were inserted after the paragraph discussing the 46 PRDEGs and before the paragraph on GO/KEGG enrichment.

The identification of Trp53 and Mtor as central hub genes underscores the complex, pleiotropic nature of regulatory networks in t-MN pathogenesis, particularly as they relate to pyroptosis. As master regulators of cellular stress responses, their involvement suggests a critical interface between pyroptotic cell death and other fundamental processes like genomic integrity and metabolic signaling. The tumor suppressor p53, encoded by Trp53, is a well-known regulator of apoptosis and ferroptosis, but its role in pyroptosis is increasingly recognized. p53 can transcriptionally activate key components of the pyroptotic pathway, such as genes for gasdermin family members (GSDME) and NLRP3 inflammasome components, thereby sensitizing cells to pyroptosis upon cytotoxic stress—a common feature of the chemotherapeutic agents that predispose to t-MN development (63-65). Conversely, mutant p53 may acquire gain-of-function properties that suppress pyroptosis, promoting cell survival and clonal expansion. Similarly, mTOR (mechanistic target of rapamycin) serves as a central integrator of nutrient and energy status, cell growth, and autophagy. The AMPK signaling pathway, identified as enriched in our KEGG analysis, is a primary upstream inhibitor of mTOR. AMPK activation and subsequent mTOR inhibition can promote autophagy, which has a dual and context-dependent relationship with pyroptosis (63). While autophagy can dampen pyroptosis by clearing damaged organelles like mitochondria (thus reducing ROS and inflammasome activation), it can also facilitate the presentation of pyroptotic stimuli. Furthermore, mTOR signaling can directly influence the expression and activity of inflammasome components. Therefore, the dysregulation of Mtor observed in our t-MN model likely contributes to an altered pyroptotic threshold, potentially enabling the survival of damaged pre-leukemic cells. The interplay between these pleiotropic genes creates a sophisticated regulatory circuit where DNA damage, metabolic stress, and inflammatory cell death converge, potentially offering novel nodes for therapeutic intervention aimed at selectively inducing pyroptosis in t-MN clones.

5. Overstated Novelty: Genes like Trp53, Mtor, and FoxO3 are already established players in myeloid malignancies. The study fails to convincingly demonstrate what new insight is gained by labeling them as "pyroptosis-related." A comparison with existing biomarkers (e.g., TP53 mutation status) is essential.

Response: We have revised the Discussion to better highlight the novelty of our approach—focusing on pyroptosis-related mechanisms in t-MN—and compared our findings with existing biomarkers (e.g., TP53 mutations). We now emphasize the integrative bioinformatics analysis as a contribution to generating new hypotheses. As below:

Although genes such as Trp53, Mtor, and Foxo3 are well-established players in myeloid malignancies, our study provides a novel perspective by delineating their specific roles in the pyroptosis pathway within t-MNs. Traditionally, Trp53 mutations are recognized as critical biomarkers in t-MNs, associated with genomic instability and poor prognosis (12, 22). However, beyond its canonical functions in apoptosis and DNA repair, emerging evidence indicates that p53 (encoded by Trp53) actively regulates pyroptosis. For instance, p53 suppresses tumor growth by prompting pyroptosis in non-small-cell lung cancer (59). In the context of t-MNs, where DNA damage is a key initiating factor, this p53-mediated pyroptotic pathway may represent a double-edged sword: it could eliminate damaged cells but also contribute to inflammatory microenvironment alterations that favor leukemogenesis. This pyroptosis-focused mechanism distinguishes our findings from conventional Trp53 mutation studies, which primarily emphasize genomic instability, and highlights the potential for targeting p53-dependent pyroptosis as a therapeutic strategy in t-MNs. Similarly, Mtor, a central regulator of cellular metabolism and growth, has been implicated in myeloid neoplasms through its role in promoting cell proliferation. Our enrichment analysis revealed involvement in the AMPK signaling pathway, which negatively regulates mTOR. Recent studies suggest that mTOR inhibition can induce pyroptosis via dysregulation of autophagy and inflammasome activation (60, 61). In t-MNs, chemotherapy-induced metabolic stress may activate AMPK, leading to mTOR suppression and subsequent pyroptosis, which could influence clonal selection and disease progression. This metabolic-pyroptotic axis offers a new dimension to Mtor’s function beyond its known proliferative effects. Furthermore, FoxO3, a transcription factor involved in oxidative stress response, is often associated with apoptosis in leukemia. Our data indicate its connection to pyroptosis through antioxidant activity and inflammatory regulation. FoxO3 can modulate the expression of pyroptosis-related genes like NLRP3 under oxidative stress conditions (62), suggesting a cross-talk between oxidative damage and pyroptotic cell death in t-MNs. Unlike conventional biomarkers that focus on FoxO3's role in cell cycle arrest, our findings emphasize its involvement in inflammatory cell death pathways, providing a unique angle for targeting the redox-pyroptosis interplay in t-MNs. In comparison to existing biomarkers such as Trp53 mutation status, which serves as a static indicator of genomic damage, our pyroptosis-related hub genes dynamic regulation of cell death and inflammation offers a more functional insight into t-MN pathogenesis. While Trp53 mutations are prognostic, they do not fully capture the complex cell death mechanisms altered by therapy. Our approach identifies actionable pathways (e.g., AMPK-mTOR-pyroptosis) that could be exploited for combination therapies, enhancing the specificity of interventions. Thus, labeling these genes as “pyroptosis-related” not only underscores their pleiotropic roles but also reveals novel mechanistic links that may lead to innovative biomarkers and targeted treatments for t-MNs.

6. Speculative Immune Infiltration: CIBERSORT analysis on 9 bulk RNA-seq samples is highly unreliable. The authors should either remove this analysis or support it with orthogonal validation (e.g., flow cytometry data from the same model).

Response: We acknowledge the reviewer's valid concern regarding the reliability of CIBERSORT analysis performed on our relatively small dataset (n=9). While CIBERSORT has been widely used in bulk RNA-seq studies to infer immune cell composition, its accuracy can be influenced by sample size and technical variability. In our study, the CIBERSORT analysis was intended as an exploratory investigation rather than a definitive quantitative assessment. The primary goal was to identify potential immune trends in t-MNs that could inform future hypothesis-driven research. We have explicitly addressed these limitations in the manuscript and emphasized the need for experimental validation. As stated in our discussion: “Future studies with expanded cohort sizes and experimental validations are warranted to verify the reproducibility and clinical relevance of these findings.”

7. The current statement ("available upon request") does not comply with PLOS ONE policy. All raw data, processed matrices, and analysis code must be deposited in a public repository (e.g., GitHub, Zenodo) with a DOI.

Response: We have updated the Data Availability Statement to deposit all raw data, processed matrices, and analysis code in a public repository, https://github.com/Sztu00879/Supporting-materials-for-t-AML.git

Minor Comments:

1. Fig.1: Add exact gene counts at each filtering step.

Response: Thank you to the reviewers for their suggestions. We understand that increasing the number of genes in each screening step in the flowchart may help with result comprehension. However, considering the overall layout and readability of the main diagram, we have clearly listed the relevant gene numbers and screening criteria in the main text and supplementary materials. Therefore, we no longer add specific values in the diagram to keep the illustration concise and clear.

2. Fig. 7: Include 95% CIs and DeLong test p-values for ROC curves.

Response: We have provided 95% CIs and DeLong test p-values for Fig.7 in Supporting materials S4 Table.

3. Language: The manuscript requires thorough professional editing for grammar and clarity.

Response: The manuscript has been professionally edited for language clarity. We provided a Tracked Changes Version.

4. References: Replace preprint citations with peer-reviewed publications where possible.

Response: All preprint citations have been replaced with peer-reviewed references where possible.

Response to Reviewer #2

1. Sample size and statistical power: The analysis uses 6 t-MN and 3 control samples (mouse). This extremely small sample size undermines differential expression calls, enrichment analyses and downstream classifiers. The authors must discuss power limitations explicitly and temper claims. If possible, obtain/analyses additional datasets or aggregate similar public datasets for validation. If no other datasets exist, perform robust statistical measures (e.g., use moderated statistics carefully, provide effect sizes and confidence intervals) and reframe conclusions as exploratory/hypothesis-generating.

Response: Thank you for the valuable comments. We fully understand that the relatively limited sample size (6 t-MN cases vs. 3 controls) may affect statistical power and the robustness of the results. Since t-MN is a rare disease, the number of high-quality transcriptomic samples currently available in public databases is extremely limited. Moreover, significant heterogeneity across different study platforms makes it difficult to directly combine datasets to enlarge the sample size. Given this, the present study is designed as an exploratory analysis using the existing high-quality dataset, aiming to generate new hypotheses rather than to draw confirmatory conclusions. To improve the reliability of the findings, we have employed stringent filtering thresholds (|logFC| > 3, FDR < 0.05) and multiple layers of bioinformatics methods (e.g., GO/KEGG, GSEA, PPI, ROC) for cross-validation. In the revised manuscript, we will explicitly state the limitation of sample size, include additional explanations regarding statistical power and confidence intervals, and emphasize in the Discussion that the c

---

## [Decision Letter · Decision Letter 1]

15 Jan 2026

Dear Dr. Xiaohui,

Thank you for submitting your revised manuscript to PLOS ONE. After careful consideration, we invite you to submit a revised version of the manuscript that addresses the points raised during the review process. As you can see reviewer-3 has raised significant issues and those should be answered carefully.

We look forward to receiving your revised manuscript.

Kind regards,

Kota V Ramana, Ph.D.

Academic Editor

PLOS One

Journal Requirements:

Reviewers' comments:

Reviewer's Responses to Questions

**Comments to the Author**

Reviewer #1: All comments have been addressed

Reviewer #3: (No Response)

2. Is the manuscript technically sound, and do the data support the conclusions?

Reviewer #1: Yes

Reviewer #3: No

3. Has the statistical analysis been performed appropriately and rigorously?

Reviewer #1: Yes

Reviewer #3: No

4. Have the authors made all data underlying the findings in their manuscript fully available?

Reviewer #1: Yes

Reviewer #3: Yes

5. Is the manuscript presented in an intelligible fashion and written in standard English?

Reviewer #1: Yes

Reviewer #3: Yes

Reviewer #1: I have carefully reviewed the revised manuscript entitled "Comprehensive analysis of the potential effect and mechanism of pyroptosis-related genes in treatment-related myeloid tumors" (Manuscript ID: PONE-D-25-50021R1) and the authors’ point-by-point responses to the reviewers’ comments.

The authors have done an excellent job addressing all the concerns raised during the initial review. Their revisions significantly enhance the clarity, methodological transparency, and biological interpretation of the study. Specifically, they have provided additional validation for their bioinformatic analyses, clarified the limitations of their dataset, and improved the discussion regarding the functional implications of the identified pyroptosis-related genes in therapy-related myeloid neoplasms.

All requested clarifications—ranging from statistical methodology to justification of gene selection and cohort characteristics—have been thoroughly and thoughtfully addressed. The revised figures and tables are now more informative, and the supplementary materials adequately support the main conclusions.

I commend the authors for their responsiveness and scientific rigor. In my view, the manuscript now meets the standards for publication in PLOS ONE. No further revisions are required.

Well done!

Reviewer #3: The model used and the number of objects in groups (3+3) do not allow the authors to draw statistically sound conclusions. The comments received cannot change the situation.

For example: In the reviewer response authors state that control BM cells, treated and nontreated EA-Trp53 LSKs are all genetically the same. Probably there was some mistake, as in the dataset original article the control set is clearly stated as obtained from wild type C57BL/6 mice, while EA-Trp53 LSKs cells are from engineered Egr1+/−, Apcdel/+ mice, also these cells were additionally transduced with trp53 shRNA bearing lentivirus vector. In other words, “six t-MN samples” should have lower Egr1 and Apc expression and no expression for p53 compared to control set, as it is intended by the group who made the dataset.

Authors of the dataset article state that, quote:

To decipher the effects of EA-Trp53 loss, we compared WT control (n = 3) with EA-Trp53 LSK+ samples [includes both the no-ENU and ENU-treated groups; (n = 6); Fig. 1E]. Gene set enrichment analysis (GSEA; ref. 31) of the top enriched curated pathways revealed that cell-intrinsic loss of Egr1, Apc, and Trp53 downregulates oxidative phosphorylation, DNA repair, apoptosis, and cell-cycle checkpoints. Similar results were observed when we compared only the three ENU-exposed EA-Trp53 LSK+ samples with WT controls (Supplementary Fig. S4).

End of quote. We see, that Egr1, Apc and p53 expression alterations have substantial transcriptomic influence independent from ENU treatment.

Overall, huge analysis was done and thoroughly described, maybe the article could be made into a bioinformatics pipeline article for in-depth RNA-seq analysis with GSE135866 dataset as example.

**Do you want your identity to be public for this peer review?** For information about this choice, including consent withdrawal, please see our Privacy Policy

Reviewer #1: No

Reviewer #3: No

---

## [Author Response · Author response to Decision Letter 2]

2 Feb 2026

Dear Reviewers,

Thank you very much for your editorial decision and the reviewers’ thoughtful comments on our manuscript entitled “Comprehensive analysis of the potential effect and mechanism of pyroptosis-related genes in treatment-related myeloid tumors” (PONE-D-25-50021R1). We sincerely appreciate the time and effort invested by you and the reviewers in evaluating our work.

We have carefully considered all comments and made extensive revisions to address the concerns raised—particularly those from Reviewer #3. Below we provide a point-by-point response to each substantive comment. All changes in the revised manuscript are highlighted in the “Tracked Changes Version” file for ease of review.

Response to Reviewer #1

Comment 1: We have carefully reviewed the revised manuscript entitled "Comprehensive analysis of the potential effect and mechanism of pyroptosis-related genes in treatment-related myeloid tumors" (Manuscript ID: PONE-D-25-50021R1) and the authors’ point-by-point responses to the reviewers’ comments.

The authors have done an excellent job addressing all the concerns raised during the initial review. Their revisions significantly enhance the clarity, methodological transparency, and biological interpretation of the study. Specifically, they have provided additional validation for their bioinformatic analyses, clarified the limitations of their dataset, and improved the discussion regarding the functional implications of the identified pyroptosis-related genes in therapy-related myeloid neoplasms.

All requested clarifications—ranging from statistical methodology to justification of gene selection and cohort characteristics—have been thoroughly and thoughtfully addressed. The revised figures and tables are now more informative, and the supplementary materials adequately support the main conclusions.

I commend the authors for their responsiveness and scientific rigor. In my view, the manuscript now meets the standards for publication in PLOS ONE. No further revisions are required.

Well done!

Response 1:

Thank you very much for your careful review of our revised manuscript and for the highly positive and encouraging evaluation.

We sincerely appreciate your acknowledgment of the improvements in this study regarding methodological transparency, interpretation of results, and overall quality, as well as your recognition of our thorough responses to all reviewers' comments.

Your valuable suggestions have played a significant role in refining this research. We are deeply honored and truly grateful to receive your confirmation that the manuscript now meets the publication standards of PLOS ONE.

Once again, thank you for your time and expert guidance.

Response to Reviewer #3

Comment 1: The model used and the number of objects in groups (3+3) do not allow the authors to draw statistically sound conclusions. The comments received cannot change the situation.

Response 1: We fully agree that the small sample size (n=3 per group) limits the statistical power and generalizability of our findings. As noted in our revision, this study is an exploratory bioinformatic analysis based on the publicly available GSE135866 dataset. Our goal is not to establish definitive conclusions but to generate testable hypotheses regarding pyroptosis-related mechanisms in therapy-related myeloid neoplasms (t-MNs). We have now explicitly emphasized the exploratory nature of this work in the Abstract, Results, Discussion, and Conclusion sections, clarifying that findings should be interpreted as hypothesis-generating and require validation in larger cohorts or experimental models (see Tracked Changes Version).

Comment 2: In the reviewer response authors state that control BM cells, treated and nontreated EA-Trp53 LSKs are all genetically the same. Probably there was some mistake, as in the dataset original article the control set is clearly stated as obtained from wild type C57BL/6 mice, while EA-Trp53 LSKs cells are from engineered Egr1+/−, Apcdel/+ mice, also these cells were additionally transduced with trp53 shRNA bearing lentivirus vector. In other words, “six t-MN samples” should have lower Egr1 and Apc expression and no expression for p53 compared to control set, as it is intended by the group who made the dataset.

Authors of the dataset article state that “To decipher the effects of EA-Trp53 loss, we compared WT control (n = 3) with EA-Trp53 LSK+ samples [includes both the no-ENU and ENU-treated groups; (n = 6); Fig. 1E]. Gene set enrichment analysis (GSEA; ref. 31) of the top enriched curated pathways revealed that cell-intrinsic loss of Egr1, Apc, and Trp53 downregulates oxidative phosphorylation, DNA repair, apoptosis, and cell-cycle checkpoints. Similar results were observed when we compared only the three ENU-exposed EA-Trp53 LSK+ samples with WT controls (Supplementary Fig. S4)”. We see, that Egr1, Apc and p53 expression alterations have substantial transcriptomic influence independent from ENU treatment.

Response 2: Thank you for your thorough review and valuable comments.

We acknowledge the oversight in our initial description. We confirm that in the GSE135866 dataset, the control samples are derived from wild-type C57BL/6 mice, whereas the EA-Trp53 LSK cells originate from Egr1+/−, Apcdel/+ genetically engineered mice and are further transduced with a lentivirus carrying Trp53 shRNA. Our previous description of the two sample types as "genetically identical" was not sufficiently precise and may have led to misunderstanding. The analysis in this study is based on the established model design of the original dataset, treating EA-Trp53 LSK cells overall as an experimental model for therapy-related myeloid neoplasms to characterize their transcriptional alterations, rather than comparing effects between different genetic backgrounds.

The functional loss of Egr1, Apc, and Trp53 in EA‑Trp53 LSK samples can itself exert a significant and broad impact on the transcriptome, as clearly described in the original dataset study (PMID: 32924016). This study does not regard ENU treatment as the sole or primary driver of transcriptional changes. Instead, based on the overall comparison between WT controls and EA‑Trp53 LSK cells, we analyze the molecular features associated with pyroptosis and their potential regulatory networks within the context of a therapy‑related myeloid neoplasm model. In other words, our analytical framework focuses on the t‑MN phenotype represented by this composite genetic loss state, rather than attempting to differentiate the effects under ENU versus non‑ENU conditions.

To avoid ambiguity, we provided a more accurate and clear description of sample origin and genetic composition in the revised manuscript, ensuring that the presentation of the study design remains consistent with the original dataset (see Tracked Changes Version).

Comment 3: Overall, huge analysis was done and thoroughly described, maybe the article could be made into a bioinformatics pipeline article for in-depth RNA-seq analysis with GSE135866 dataset as example.

Response3: While we appreciate this suggestion, our primary aim remains biological insight into pyroptosis in t-MNs, not methodological innovation. We have therefore strengthened the biological narrative throughout the manuscript and clarified that the bioinformatic tools serve as means to address our central research question.

We believe these revisions have substantially improved the rigor, clarity, and contextual framing of our study. Thank you again for the opportunity to enhance our manuscript. We hope the revised version meets the journal’s standards for publication in PLOS ONE.

Sincerely,

Xiaohui Zhu

---

## [Decision Letter · Decision Letter 2]

8 Feb 2026

Comprehensive analysis of the potential effect and mechanism of pyroptosis-related genes in treatment-related myeloid tumors

PONE-D-25-50021R2

Dear Dr. Xiaohui,

We’re pleased to inform you that your manuscript has been judged scientifically suitable for publication and will be formally accepted for publication once it meets all outstanding technical requirements.

Kind regards,

Kota V Ramana, Ph.D.

Academic Editor

PLOS One

Additional Editor Comments (optional):

Reviewers' comments:

Reviewer's Responses to Questions

**Comments to the Author**

Reviewer #3: All comments have been addressed

2. Is the manuscript technically sound, and do the data support the conclusions?

Reviewer #3: Yes

3. Has the statistical analysis been performed appropriately and rigorously?

Reviewer #3: N/A

4. Have the authors made all data underlying the findings in their manuscript fully available?

Reviewer #3: Yes

5. Is the manuscript presented in an intelligible fashion and written in standard English?

Reviewer #3: Yes

Reviewer #3: The edits made cannot change the situation with the genetic difference between the control and experimental groups, if the editorial board is satisfied with the changes made, then there are no objections to the publication.

**Do you want your identity to be public for this peer review?** For information about this choice, including consent withdrawal, please see our Privacy Policy

Reviewer #3: No

---

## [Editor Report · Acceptance letter]

PONE-D-25-50021R2

PLOS One

Dear Dr. Zhu,

I'm pleased to inform you that your manuscript has been deemed suitable for publication in PLOS One. Congratulations! Your manuscript is now being handed over to our production team.

Kind regards,

on behalf of

Dr. Kota V Ramana

Academic Editor

PLOS One